



# Observational study for strong downslope wind event under fine weather
# condition during ICE-POP 2018

**Chia-Lun Tsai[1], Kwonil Kim[1], Yu-Chieng Liou[2], Jung-Hoon Kim[3] , YongHee Lee[4], and**
**GyuWon Lee\*[1]**
[1]Department of Astronomy and Atmospheric Sciences, Center for Atmospheric REmote
sensing (CARE), Kyungpook National University, Daegu, Korea
[2]Department of Atmospheric Sciences, National Central University, Jhongli, Taiwan
[3]School of Earth and Environmental Sciences, Seoul National University, Seoul, Korea
[4]Numerical Modeling Center (NMC), Korea Meteorological Administration, Seoul, Korea

\* Corresponding author: Prof. GyuWon Lee, E-mail: gyuwon@knu.ac.kr





## Abstract

A strong downslope wind event under fine weather condition on 13–15 February 2018 was examined by various observational and high resolution reanalysis datasets during the 2018 Winter Olympic and Paralympic games in Pyeongchang, Korea. High spatio-temporal resolution of wind information was obtained by Doppler lidars, automatic weather stations (AWS), wind profiler, and sounding observations under the International Collaborative Experiments for Pyeongchang 2018 Olympic and Paralympic winter games (ICE-POP 2018). This study aimed to understand the possible generation mechanisms of localized strong wind event across high mountainous areas and in the lee side of mountains associated with the underlying large-scale pattern of a low-pressure system (LPS). The spatial distribution of linear trends for surface wind shows different patterns, exhibiting increased trend in the lee side and a persistent one in mountainous areas with the approaching LPS. Surface wind speed was intensified dramatically from ~3 to ~12 m s$^{-1}$ (gust was stronger than 20 m s$^{-1}$ above ground) at a surface station in the lee side (named as GWW). However, the mountainous station at DGW site appeared to have a persistently strong wind (~10 m s$^{-1}$) during the research period. Budget analysis of horizontal momentum equation and local reanalysis data suggests that the pressure gradient force (PGF) derived by adiabatic warming along the downslope and subsequent hydraulic jump in the lee side of mountains was a main factor in the acceleration of the surface wind at the GWW site. Detailed analysis of the retrieved 3D winds reveals that the PGF also dominate at the DGW site, which causes the persistent strong wind that is related to the channeling effect across the valley areas in the mountain range. The observational evidence presented here shows that the different mechanisms in local areas under the same synoptic condition with LPS are important references in determining the strength and persistence of the orographic-induced strong winds under fine weather condition.





## 1 Introduction


In mountain regions, wind is an important atmospheric phenomena as enhanced precipitation
may be usually caused by the wind impinging topography (Medina et al., 2007; Yu and Cheng,
2008; Panziera and Germann, 2010; Houze, 2012; Yu and Tsai, 2017; Tsai et al., 2018).
Therefore, topography could significantly affect the behavior of winds to accelerate the wind
speed or to change the wind direction. Such orographically strong wind and mountain waves can
highly induce huge impacts on aviation operations (Clark et al., 2000; Kim and Chun, 2010, 2011;
Kim et al., 2019; Park et al., 2016, 2019), outdoor sport activities, and forest wildfires in a
relatively drier environment under the fine weather conditions (Smith et al., 2018). Downslope
windstorm can produce strong wind in the lee side and plays an essential role in creating and
maintaining the wildfires near the northern California with the easterly winds across Sierra
Nevada and the southern Cascade Mountains (Mass and Ovens, 2019). Lee et al. (2020) also
suggested that the downslope windstorm favored the wildfires along the northeastern coast of
Korea with the westerly winds across the Taebeak Mountain Range (TMR). The wind speed was
also usually accelerated locally near the narrow valley or channel in between the mountains like
the "gap wind" occurring along the strait of Juan de Fuca (Reed, 1931; Colle and Mass, 2000) in
Washington, Columbia River Gorge in Oregon (Sharp, 2002), and Jangjeon area in South Korea
(Lee et al., 2020).
Large-scale environmental conditions are key factors in determining the location where the
downslope windstorm will be generated. The downslope windstorm usually occurs at the lee side,
and the upstream prevailing wind direction is mostly perpendicular to the orientation of the
mountain range. The elevated inversion-layer and the height of mean-state critical-level are also
important references to evaluate the occurrence of downslope windstorm. The mechanisms of the
downslope windstorm could usually be explained by hydraulic jump, partial reflection, and
critical-level reflection from various numerical and theoretical studies in the past few decades



(Long, 1953; Houghton and Kasahara, 1968; Klemp and Lilly, 1975; Smith, 1985; Durran, 1990;
Afanasyev and Peltier, 1998; Epifanio and Qian, 2008; Rögnvaldsson et al., 2011; Cao and
Fovell, 2016). The combination of hydraulic jump and wave breaking can also enhance the
downslope windstorm and increase the wind speed (Shestakova et al., 2018; Tollinger et al.,
2019). The pressure gradient force (PGF) is one of the possible factors that accelerate the wind
speed near the exit of gap between the mountains (Reed, 1931; Finnigan et al., 1994; Colle and
Mass, 2000).

A few previous studies provided insightful explanations in numerical aspect about the

development of the strong wind associated with the downslope windstorm along the northeastern
coast of South Korea (in the lee side of the Taebaek Mountain Ranges; TMR). Most strong
downslope wind events can be mainly explained by the three mechanisms in this region: hydraulic
jump, partial reflection, and critical-level reflection (Lee, 2003; Kim and Cheong, 2006; Jang and
Chun, 2008; Lee and In, 2009). The strong wind possibly occurs in any season with appropriate
environmental conditions such as westerly and upstream inversion. Lee et al. (2020) confirms
these conclusions with numerical modeling studies. Furthermore, they also found that PGF is one
of the possible factors to cause the gap wind, and the variability of PGF is highly related to the
localized topographic features. However, it still lacks sufficient observational evidence to
examine the strong downslope wind events in detail because relatively dense wind observations
from ground-based remote sensing techniques could not be easily collected under fine weather
conditions.

Pyeongchang hosted the Winter Olympic and Paralympic Games in 2018 (most venues are

located at coastal and higher altitude areas of the TMR). More detailed weather conditions and
accurate prediction for several key parameters such as precipitation, visibility, wind directions,
and wind speed are very important to ensure the safety of all game players and attendees. The
Numerical Modeling Center (NWC), Korea Meteorological Administration (KMA) organized an
intensive field experiment named the International Collaborative Experiments for Pyeongchang



2018    Olympic    and    Paralympic    winter    games,    ICE-POP    2018
(http://155.230.157.230:8080/Icepop_2018/index.jsp). A very dense observational network was
built to provide a good quality observational dataset in high temporal and spatial resolution either
under precipitation or fine weather conditions. Many kinds of instruments were involved in the
ICE-POP 2018 that allows the investigation of the nature of the strong wind event nearby
mountainous areas in observational aspects.

Wider coverage of wind information can be usually obtained by the Doppler radar; however,

regular meteorological Doppler radars cannot detect radial winds in fine weather conditions.
Scanning Doppler lidar can be the best solution in obtaining a more complete wind information
in such conditions even with finer resolutions. Using the Doppler lidar to document the
downslope flow and rotors, Kühnlein et al. (2013) found that the transient internal hydraulic jump
is characterized by turbulence. Menke et al. (2019) identified the recirculation zone over a
complex terrain using six scanning Doppler lidars. The interactions between the winds and terrain
essentially affected the occurrence of flow recirculation. However, only radial winds were used
resulting in incomplete wind observations that can only provide very limited information for the
realistic airflow structures. Complete 3D wind fields could be retrieved from 4DVAR (4D-
Variational Assimilation) using Doppler lidar. The accuracy of wind speed, direction and water
vapor flux will be improved when assimilating the lidar data (Kawabata, 2014). Thus, lidar
observations can indeed provide a good quality of 3D wind information under fine weather
condition.

The objective of this study is to use high spatiotemporal resolution of 3D winds and

observational data to investigate the fine-scale structural evolution of strong downslope winds
over the complex terrain in northeastern part of South Korea (i.e., in the Pyeongchang area )
during the period 13–15 February 2018. Multiple Doppler lidars, automatic weather station
(AWS), wind profiler, sounding observations, and local reanalysis (LDAPS: Local Data
Assimilation and Prediction System) datasets were adopted to derive detailed 3D wind fields over



TMR and northeastern coastal regions through the WISSDOM (WInd Synthesis System using
DOppler Measurements, Liou and Chang, 2009; Tsai et al., 2018) synthesis. In particular, this
study tries to recognize the mechanisms of the strong wind over the mountainous area and in the
lee side areas of TMR while a LPS passing through the northern side of the Korean peninsula. A
strong downslope wind case was selected for the further analysis in this study not only because
the Olympic games were interrupted due to the strong wind but also because dense observations
are available during the ICE-POP 2018. Interactions between the complex terrain and large-scale
airflow of the LPS could be adequately explored in this study. Furthermore, three scanning
Doppler lidars were established in this area, which provided more sufficient wind information
under fine weather condition.

**2 Data and methodology**
**2.1 Scanning Doppler lidar**

Two different models of scanning Doppler lidars were adopted in this study: (1) "WINDEX-

2000" produced by the manufacturer Laser Systems and (2) the "Stream Line" produced by the
manufacturer HALO Photonics. The scanning Doppler lidar can measure the radial Doppler
velocity by detecting atmospheric aerosols and dusts via laser (class 1M) in a very high spatial
resolution. The radial winds were sufficiently observed by adjustable scanning strategy in three
modes: Plan Position Indicator (PPI), Range Height Indicator (RHI), and Zenith pointing (ZP).
Furthermore, the complete wind information could be constructed using these lidar observations
under fine weather condition via WISSDOM.

The WINDEX-2000 lidar operated a full volume scan every ~27 min in which seven PPIs

(the elevation angles of 5°, 7°, 10°, 15°, 30°, 45°, and 80°) and one hemispheric RHI (the azimuth
angle of 0°, that is, started from the north). There are 344 gates along a radar radial direction with



360 azimuth angles between 0º and 360º. The gate spacing is 40 m and the maximum observed
radius distance is ~13 km. The Stream Line lidar operated a full volume scan every ~13 min in
which five PPIs (7º, 15º, 30º, 45º, and 80º before 10:00 UTC on 14 Feb. 2018 and 4º, 8º, 14º, 25º,
and 80º after 10:00 UTC) and two hemispheric RHIs (the azimuth angle of 51º and 330 º). There
are 1660 gates along the 360 radar beams with azimuth angles between 0º and 360º. The gate
spacing is 60 m and the maximum observed radius distance can reach to ~100 km.

Quality control (QC) of the radial winds (in PPI and RHI modes) was done by applying

SNR (signal noise ratio) threshold in advance. To obtain correct and useful measurements, QC is
necessary for each lidar observation, where the non-meteorological echoes are removed as the
threshold of SNR is smaller than 0.04.
**2.2 Surface weather station, sounding, wind profiler and LDAPS**

Figure 1 shows an observational domain and domain of mountain clusters during the ICE-

POP 2018. AWS mainly provides the surface wind speed, wind direction, pressure, and
temperature in high spatiotemporal resolution (1-min interval and ~10 km distance between each
AWS station). In addition to the regular KMA AWS, additional AWS sites were deployed in the
mountainous area during the ICE-POP 2018. Thus, dense AWS network (black dots in Fig. 1b)
can be utilized to document detailed evolutions of surface parameters and could also be
observational inputs for WISSDOM (the details is Section 2.3). Five soundings are launched
every 3 hours and the sounding sites are all located in the studied domain near the northeastern
part of South Korea (black square in Fig. 1a). Such dense sounding observations provide
environmental winds, temperature profiles, and stability information in a very fine-scale across
the mountainous and coastal areas. The wind profiler is deployed at the GWW site to measure
the winds in case of lacking sounding observations. In addition, the high temporal resolution of
wind profiler measurements (10-min interval) could potentially be reference for the surface and



retrieved winds.

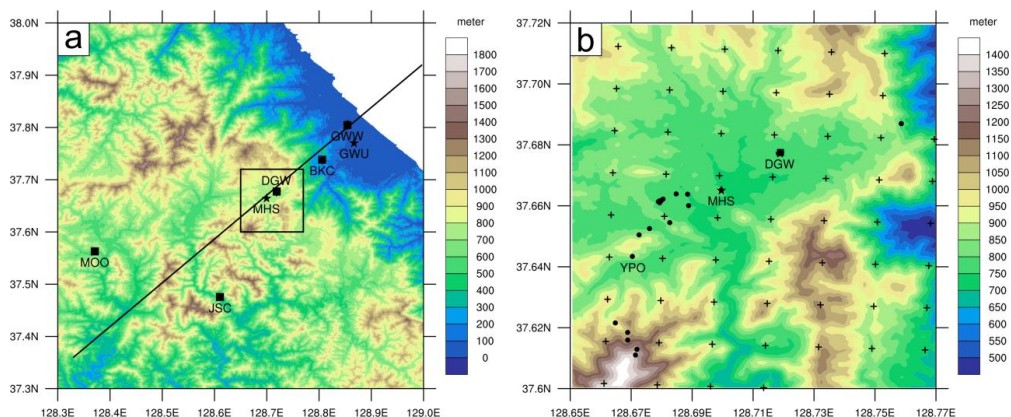


**Figure 1.** (a) Observations used in this study and the topographic features from the DEM (digital elevation model)
in the northeastern Korea (corresponding to the bigger box in the Fig. 3) and (b) WISSDOM synthesis domain
adopted in this study corresponding to the box in Fig. 1a. The locations of the scanning Doppler lidar sites are denoted
by asterisks. The locations of the sounding sites are denoted by squares. Note that the sounding and lidar observations
are both operated at DGW site and a wind profiler is located at GWW site. The locations of the AWS sites and
LDAPS grids are denoted by dots and plus symbols, respectively.

The LDAPS is a 3DVAR numerical weather prediction (NWP) product generated from
KMA with a spatial (temporal) resolution of ~1.5 km (3 hours) with 70 levels in vertical. High
resolution 3D wind fields from LDAPS will be one of the "observational" inputs in WISSDOM.
This dataset is freely available from the KMA web site (https://data.kma.go.kr). Note that a
relatively weaker weighting of the LDAPS input will be set in WISSDOM because this study
prefers to emphasize the contribution from the observations.
**2.3 WInd Synthesis System using DOppler Measurements (WISSDOM)**
WISSDOM was originally developed by Liou and Chang (2009) and has been applied in the
Pyeongchang area (Tsai et al., 2018). This study adopted a newly improved version, which
includes more observations with more constraints compared with a previous version. In new
version of WISSDOM, the following cost function [eq. (1)] was minimized by using
mathematical variational-based method at the retrieval time,

$$J = \sum_{M=1}^{8} J_M.$$    (1)

This cost function comprised of eight constraints, and the 3D wind fields were obtained by
variationally adjusting solutions to simultaneously satisfy those constraints at the same time. The
first constraint is the geometric relation between the radial velocity $(V_r)$ observations from
multiple lidars and Cartesian winds $V_t = (u_t, v_t, w_t)$, control variables, defined as

$$J_1 = \sum_{t=1}^{2} \sum_{x,y,z} \sum_{i=1}^{N} \alpha_{1,i} \left(T_{1,i,t}\right)^2,$$    (2a)

$$T_{1,i,t} = (V_r)_{i,t} - \frac{(x - P_x^i)}{r_i} u_t - \frac{(y - P_y^i)}{r_i} v_t - \frac{(z - P_z^i)}{r_i}(w_t - W_{T,t}), \text{and}$$    (2b)

$$r_i = \sqrt{\left(x - P_x^i\right)^2 + \left(y - P_y^i\right)^2 + (z - P_z^i)^2}.$$    (2c)

Any numbers of lidar [subscripts $i$ in eq. (2a)] can be applied to this constraint at two time levels
(subscripts $t$). $\alpha_1$ in eq. (2a) is the weighting coefficient corresponding to $J_1$ (same as the
following equations for $J_2$ - $J_8$). The subscripts $i$ and $t$ in the $(V_r)_{i,t}$ represent the radial
velocity observed by the $i$-th lidar, and $(u_t, v_t, w_t)$ indicate the 3D wind at location $(x, y, z)$,
and the terminal velocity $(W_{T,t})$ of particles were estimated by radar reflectivity at two time
levels. $\left(P_x^i, P_y^i, P_z^i\right)$ are the coordinates of the $i$-th lidar, and the distance between each grid point
and the $i$-th lidar were denoted by $r_i$. Note that $W_{T,t}$ will be zero when there is no radar
reflectivity, or the terminal velocity is possibly negligible under fine weather condition.
Next constraint is the difference between the $\mathbf{V}_t$ and the background winds $(\mathbf{V}_{B,t})$ defined
in eq. (3);

$$J_2 = \sum_{t=1}^{2} \sum_{x,y,z} \alpha_2 \left(\mathbf{V}_t - \mathbf{V}_{B,t}\right)^2,$$    (3)

the sounding observations are used to be the background winds in eq. (3). The constraint of the
anelastic continuity equation is



$$J_3 = \sum_{t=1}^{2} \sum_{x,y,z} \alpha_3 \left[ \frac{\partial(\rho_0 u_t)}{\partial x} + \frac{\partial(\rho_0 v_t)}{\partial y} + \frac{\partial(\rho_0 w_t)}{\partial z} \right]^2, \qquad (4)$$

where $\rho_0$ is the air density. The fourth constraint was deduced from the vertical vorticity
equation given by
$$J_4 = \sum_{x,y,z} \alpha_4 \left\{ \frac{\partial \xi}{\partial t} + \overline{\left[ u\frac{\partial \xi}{\partial x} + v\frac{\partial \xi}{\partial y} + w\frac{\partial \xi}{\partial z} + (\xi + f)\left(\frac{\partial u}{\partial x} + \frac{\partial v}{\partial y}\right) + \left(\frac{\partial w}{\partial x}\frac{\partial v}{\partial y} - \frac{\partial w}{\partial y}\frac{\partial u}{\partial z}\right) \right]} \right\}^2, \quad (5)$$

where $f$ indicate the Coriolis parameter and the meaning of overbar in eq. (5) is the temporal
average of the two time levels. The constraint about Laplacian smoothing filter:
$$J_5 = \sum_{t=1}^{2} \sum_{x,y,z} \alpha_5 [\nabla^2 (u_t + v_t + w_t)]^2. \qquad (6)$$

232   The horizontal winds observed by the sounding, AWS and LDAPS, can be interpolated to

each given grid in the WISSDOM synthesis domain. The sixth constraint is the difference
between the $V_t$ and the sounding observations ($V_{S,t}$), as defined in eq. (7);
$$J_6 = \sum_{t=1}^{2} \sum_{x,y,z} \alpha_6 (V_t - V_{S,t})^2. \qquad (7)$$

The seventh constraint represents the discrepancy between the surface winds and AWS ($V_{A,t}$),
as expressed in eq. (8);
$$J_7 = \sum_{t=1}^{2} \sum_{x,y,z} \alpha_7 (V_t - V_{A,t})^2. \qquad (8)$$

239   Finally, the eighth constraint measures the squared errors between the horizontal winds and the

LDAPS ($V_{L,t}$), as defined in eq. (9);
$$J_8 = \sum_{t=1}^{2} \sum_{x,y,z} \alpha_8 (V_t - V_{L,t})^2. \qquad (9)$$

242   Original version of WISSDOM is just only included first five constraints, it had already applied

to synthesize high-quality 3D wind field in some of previous studies (Liou and Chang, 2009;




Liou et al., 2012, 2013, 2014, 2016; Lee et al., 2017). The primary advantages and additional
details of WISSDOM can be referred to Tsai et al. (2018). The main improvement of the new
version of WISSDOM is that all available wind observations are considered to be one of the
constraints to minimize the cost function. In addition, this new version extends its applicability
by including multiple-lidar observations and thus, realistic wind fields would be retrieved under
fine weather condition.
**3 Overview of the strong wind case**
**3.1 Synoptic condition**

Hourly ERA5 dataset was used herein to document the synoptic condition. The ERA5 data is

an atmospheric reanalysis of the globe climate and was generated by the European Centre for
Medium-Range Weather Forecasts (ECMWF, DOI: 10.24381/cds.adbb2d47). In the beginning
of the research period at 12:00 UTC on 13 February 2018, a high-pressure system (HPS) was
located in the southernmost of the Korean peninsula (as Fig. 2a). Surface southwesterly winds
were dominant from the Yellow sea to the western coast of South Korea associated with anti-
cyclonic circulation of the HPS. This stronger southwesterly wind was also related to the cyclonic
circulation of a low-pressure system (LPS) centered at 39ºN, 117ºE near Beijing, China.
Compared to the western coast, relatively weak wind existed over the land and eastern coast of
Korea. The westerly wind came from China accompanying warm air at a higher layer (850 hPa,
Fig. 2b). This veering wind also indicated that the prevailing southwesterly wind was dominated
by the warm advection. Thus, the temperature gradient existed between the land, western and
eastern coast (exceeding ~4ºK difference).

**Figure 2.** (Left panels) Horizontal winds (vectors) and surface pressure (hPa, color shading) at the surface level obtained from the ERA5 reanalysis dataset at (a) 12:00 UTC on 13 Feb., (c) 03:00 UTC on 14 Feb. and (e) 00:00 UTC on 15 Feb. 2018. (Right panels) Horizontal winds (vectors) and temperature (°K, color shading) at 850 hPa level from the same reanalysis at (b) 12:00 UTC on 13 Feb., (d) 03:00 UTC on 14 Feb. and (f) 00:00 UTC on 15 Feb. 2018. The locations of low (high) pressure systems are marked by "L" ("H").



Consequently, the LPS and HPS were both moving eastward. The surface wind became
stronger and turned to westerly over the Korean peninsula associated with the confluences
between these two systems (Fig. 2c). Horizontal pressure gradient was intensified along the
northeastern coast of Korea as the LPS moved to the East Sea. A relatively lower temperature
was detected over the mountain area (i.e., near northeastern coast of South Korea) even when the
warm advection was approaching Korea (Fig. 2d). The other HPS came from China at around
00:00 UTC on 15 February 2018 and the environmental winds surrounding Korea were switched
to relatively weak northerly winds. Relatively weaker pressure gradient and smaller temperature
differences between the western and eastern coasts were shown in Figs. 2e and 2f. No
precipitation existed along the northeastern coast of South Korea according to the AWS
observations during the research period (not show).
The research period can be separated into two stages based on the synoptic conditions as the
LPS was approaching (between 18:00 UTC on 13 February and 09:00 UTC on 14 February 2018)
and leaving (09:00 UTC on 14 February and 00:00 UTC on 15 February 2018) from Korea.
Spatial distribution of linear trends (percentage h$^{-1}$, Chou et al., 2013) of surface wind speed for
these two stages were shown in Fig. 3. Generally, the surface wind speed was increased by ~2–8
percentage h$^{-1}$ while the LPS was approaching Korea and higher increasing trend (over than 10
percentage h$^{-1}$) was related to the upstream of mountain ranges and the east coast (lee side of the
TMR) in Korea (Fig. 3a). In addition, the strong positive spatial gradient appeared from the TMR
to the east coast along with negative trends over the mountain ranges as described in the Fig. 3a.
That is, the surface wind speed slightly decreased or nearly constant (~10 m s$^{-1}$) over the TMR
while the LPS was approaching. The surface wind speed dramatically increased with the
approaching LPS. When the LPS was moving away from the east coast, the negative trend is
dramatic, in particular, over the TMR and east coastal regions (Fig. 3b).

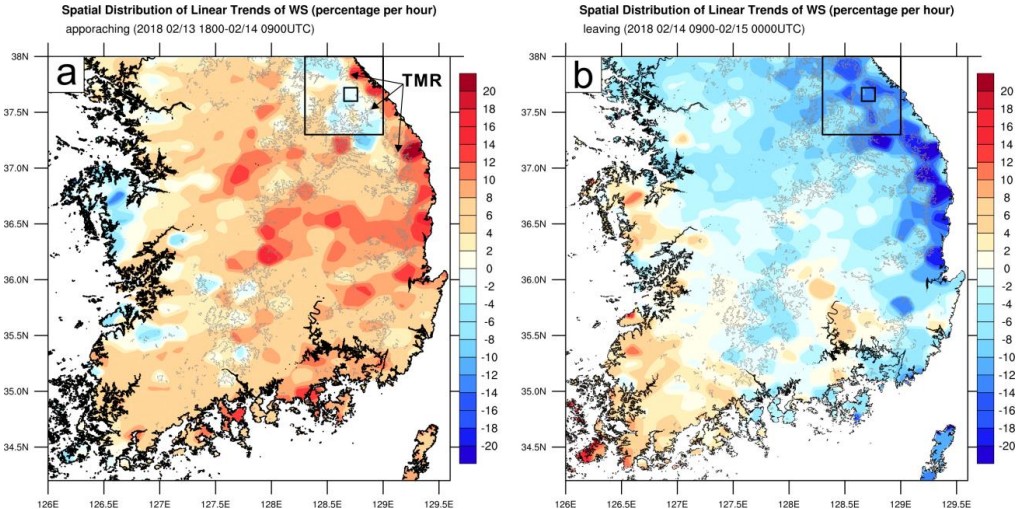

**Figure 3.** (a) Spatial distribution of linear trends of surface wind speed (unit: percentage h$^{-1}$, color shading) calculated from the AWS (automatic weather station) observations during the low-pressure system (LPS) approaching from 18:00 UTC on 13 to 09:00 UTC on 14 Feb. 2018. (b) as in (a) but for the time period when the LPS was leaving from 09:00 UTC on 14 to 00:00 UTC on 15 Feb. 2018. The topographic height thresholds of 600 m MSL (mean sea level)is indicated by gray lines and the arrows marked the location of the TMR.

## 3.2. Environmental conditions in the local area near northeastern Korea

Because the spatial distribution of linear trends of surface wind speed revealed obvious gradient just near the mountainous area and in the lee side of the TMR, two domains were selected in this study that are shown as boxes in Fig. 3. All available intensive observations are also marked in Figs. 1a and 1b, corresponding to the bigger and smaller boxes in Fig. 3. Three scanning Doppler lidars were deployed at DGW, GWU (Stream Line), and MHS (WINDEX-2000), indicated by asterisks in Fig. 1a. Five sounding stations aligned from the mountainous area to the coastal area (i.e., perpendicular with the orientation of TMR). The sounding stations MOD and JSC were located at the southwestern TMR with gentle slope and the station DGW was the closest site to most outdoor venues of the Olympic games near the crest of TMR. Station BKC and GWW were located at the northeastern slope of TMR and the coast area, respectively (as Fig. 1a). In addition, a wind profiler is located at the lee side (GWW). The WISSDOM





synthesis domain was set over the mountainous area with a horizontal spatial coverage of 12 ×
12 km² as shown in Fig. 1b. The horizontal and vertical grid size were both set to 50 m, and
vertical extension is from 0- km to 3- km height MSL (mean sea level). Additional AWS stations
were deployed around the venues (black dots in Fig. 1b) during the ICE-POP 2018. Furthermore,
the location of the LDAPS reanalysis dataset was also marked by the plus symbols.

Figure. 4 shows the variations in the environmental winds observed by the soundings and/or

wind profiler along the crossline (black line in Fig. 1a) from the mountainous area to the lee side.
Instead of lacking sounding observations at GWW site, the wind profiler observation is used to
provide the wind information near the coastal area when the LPS was passing Korea. In the
beginning of the research period, prevailing westerly winds are dominant at all sounding sites
(Fig. 4a). However, a relatively stronger wind was only measured in heights below ~1.5 km 184
at the DGW site near the crest of the TMR (~25 m s⁻¹), and weaker wind (<15 m s⁻¹) were
observed at other sites (MOO, BKC, and GWW) in lower layers on both windward slope and lee
side. The wind direction still revealed westerly at 03:00 UTC on 14 February 2018 (as Fig. 4b).
However, strong winds were detected at the TMR (DGW) and in the downslope with a wind
speed larger than 20 m s⁻¹ above the BKC and GWW sites. Although the wind speed becomes
stronger above 1.5 km MSL over the DGW site, it did not exhibit a significant change near the
surface. The results demonstrated that persistent strong winds existed over the mountainous area
(i.e., near the DGW site) while the LPS was approaching. After the LPS passed Korea, all
sounding sites exhibited the wind speed weakened near the surface (Fig. 4c).



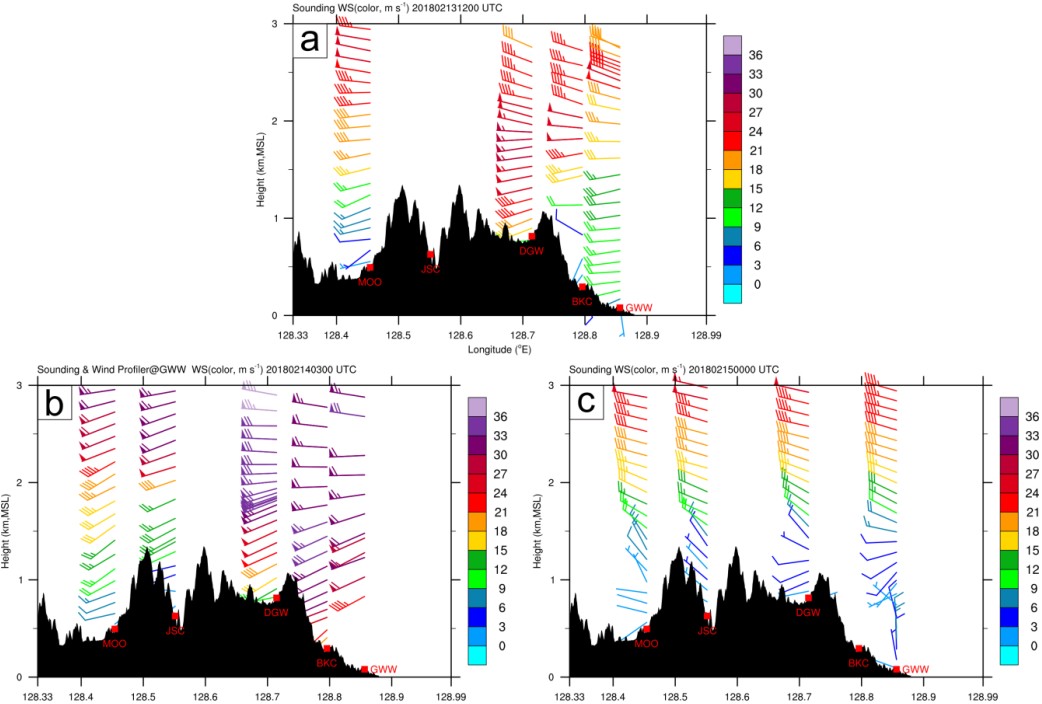

**Figure 4.** Horizontal winds observed by sounding and wind profiler along the cross line corresponding to Fig. 1a at (a) 12:00 UTC on 13 Feb. 2018, (b) 03:00 UTC on 14 Feb., and (c) 00:00 UTC on 15 Feb. 2018. A full wind barb corresponds to 5 m s⁻¹; a half barb corresponds to 2.5 m s⁻¹. The color indicates the wind speed correspond to bars. The black shading in the lower portion indicates the averaged topography along the line in Fig. 1a.

Detailed environmental conditions in the upstream of the lee site (or the TMR area) was investigated by the sounding observations at the DGW site (Fig. 5). An inversion layer existed around 800 hPa height mainly due to the warm advection accompanied by the southwesterly at 850 hPa ahead of the LPS (Figs. 2a and 2b at 12:00 UTC on 13 Feb. 2018) until it was passing through the Korea peninsula (at 03:00 UTC on 14 Feb. 2018). The temperature was increasing near the surface and became drier above the inversion layer between the two time steps. The wind direction comprised of westerly at all levels while the LPS was passing through. The wind speed became stronger above the inversion layer, but it exhibited no clear changes below ~800 hPa. It is worth to mention that the inversion layer was mainly developed by two contributions: (1) large-scale warm advection and (2) stable boundary layer. Their separate contributions would require



a modeling study for this case in the future. The sounding observations showed a good condition
for generating hydraulic jump and downslope windstorm in the lee side (Lee et al. 2020).
In summary, the environmental winds were mostly westerly in this case. However, the wind
speeds revealed different characteristics at mountainous area and lee side. Strong winds (~10 m
$s^{-1}$) persist near the surface on the mountainous area and the wind speed dramatically increased
in the lee side.

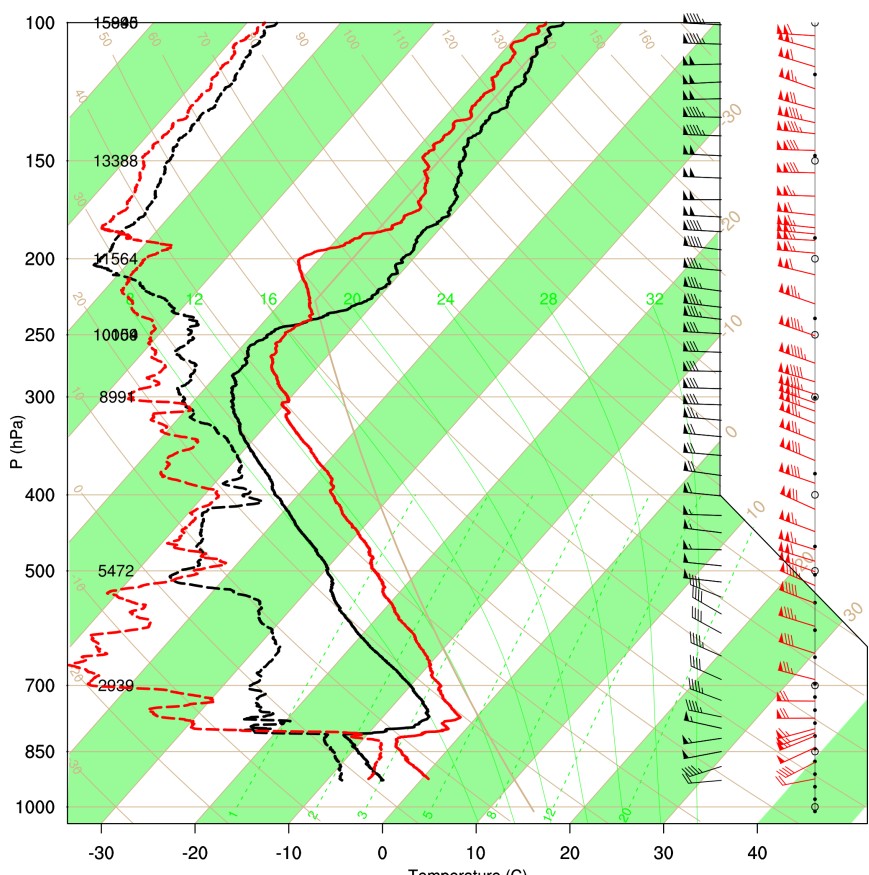


**Figure 5.** Profiles of temperature (solid lines), dew point (dashed lines), and horizontal winds observed by sounding
at the DGW site at 12:00 UTC on 13 Feb. (black lines) and 03:00 UTC on 14 Feb. (red lines) 2018. A full wind barb
corresponds to 5 m $s^{-1}$ and a half barb corresponds to 2.5 m $s^{-1}$.



## 4 Stronger winds in the lee side

### 4.1 The lidar and AWS observations

Although the environmental wind direction likely westerly, the wind speed had a dramatic increasing trend in the lee side of the TMR. The detailed wind speed and surface fluctuations were documented by lidar quasi-vertical profile (QVP, Ryzhkov et al., 2016) at GWU site (upper panel) and the AWS observations at GWW site (lower panel) as shown in Fig. 6. The wind speed was relatively weak around 6- 9 m s$^{-1}$ at the lowest layer in the beginning of the research period. Strong winds were then measured by the lidar QVP reaching ~36 m s$^{-1}$ up to ~1.5 km MSL after 00:00 UTC on 14 February 2018 (as Fig. 6a). It was clear that strong upper winds propagate toward the lower layer, intensify the wind speed near surface between 03:00 UTC and 09:00 UTC. Finally, the wind speed became weak after 09:00 UTC on 14 February. Winds observed from the sounding and wind profiler were consistent with these QVP winds (cf. Fig. 4).

Fluctuations in wind speed, direction, perturbation pressure and perturbation temperature at GWW site were shown in Fig. 6b. The changes of wind speed were similar to the lowest layer of lidar observations (cf. Fig. 6a). Relatively weak winds were measured at the early stage of the period and the surface wind speed was intensified dramatically exceeding ~12 m s$^{-1}$ between 00:00 and 06:00 UTC on 14 February (named as speed-up stage and highlighted by shaded area in the Fig. 6). The surface wind direction also showed similar patterns to the lidar observations as it had minor changes from more southerly to westerly. Although these two stations were at different locations along the northeastern coast of Korea, they revealed consistent changes on wind fields. The results also implied that the wind fields along the coast and in the lee side of the TMR have almost the same characteristics, which could be verified by the linear trend distribution of the surface wind speed (cf. Fig. 3a). Negative perturbation temperature was measured within the first 12 hours in the beginning of the period and increased after 00:00 UTC on 14 February





from ~ −3 ℃ to 4 ℃. The fluctuation of perturbation pressure showed an opposite phase with the
perturbation temperature with the changing magnitude from around 5 to −6 hPa. The wind speed
was increasing just after the perturbation temperature (pressure) was rising (dropping). That is, a
significant lag between T' (P') and wind speed is evident. Their specific relationships and
mechanisms will be clarified through a more detailed analysis in Section 4.2.

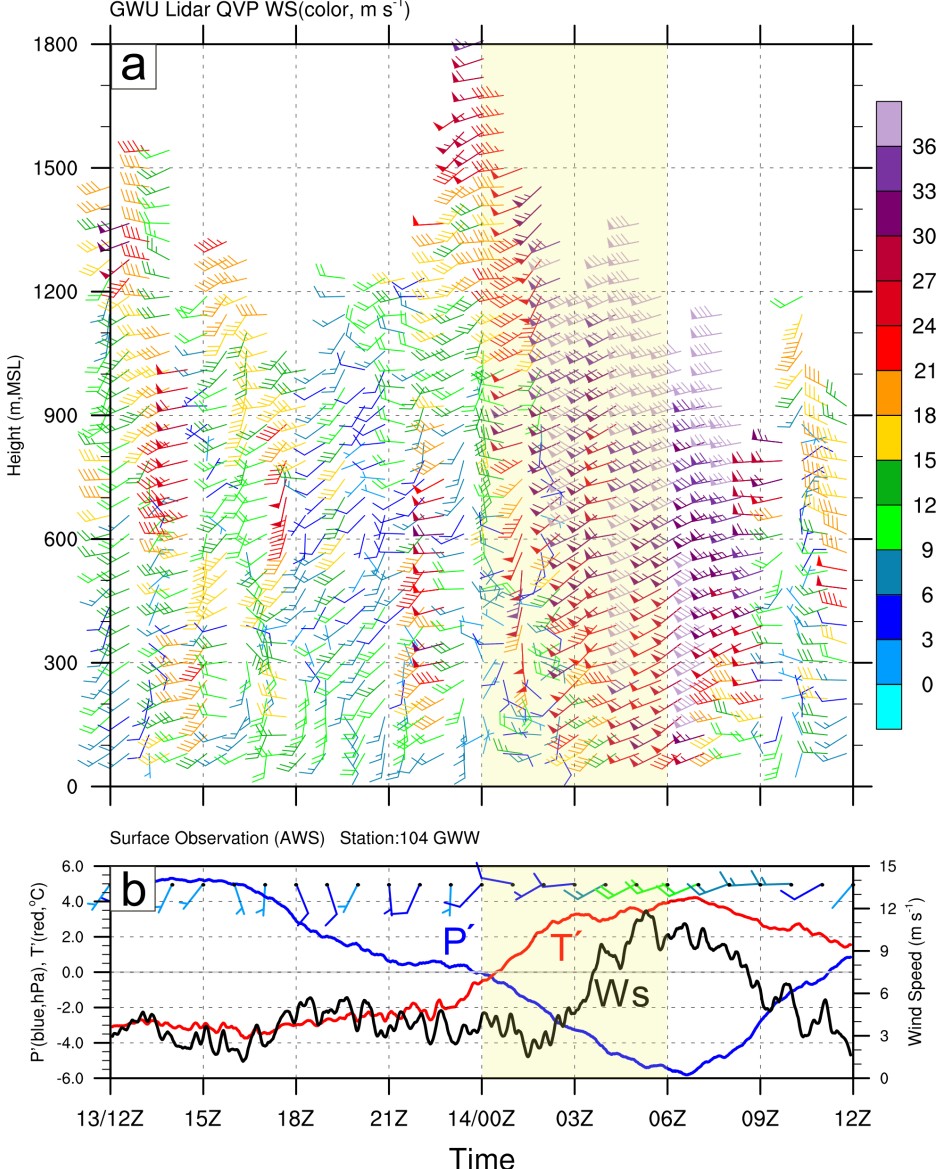




**Figure 6.** (a) Time series of QVP (quasi-vertical profile) from lidar observations at GWU site during 12:00 UTC on 13 Feb. to 12:00 UTC on 14 Feb. 2018. A full wind barb corresponds to 5 m s$^{-1}$; a half barb corresponds to 2.5 m s$^{-1}$ and the color indicates the wind speed (m s$^{-1}$) correspond to color scale. (b) Time series of horizontal winds (wind barbs), wind speed (m s$^{-1}$, black line), perturbation pressure (hPa, P', blue line) and perturbation temperature (°C, T', red line) observed from the AWS at GWW site. The time period with accelerating wind speed is also highlighted by light yellow shading (i.e., speed-up stage).

**4.2 Possible mechanism of strong winds in the lee side**

The winds can usually be accelerated by the PGF between the two sites as the stronger wind usually occurred at the site where lower pressure was located. Therefore, the DGW site was selected as the upstream from the GWW site, and the differences between the surface temperature and sea level pressure will be analyzed. A relatively warm environment was presented in the lee side of the TMR and the temperature between the DGW and GWW sites suddenly increases from ~7 °C in the beginning of the research period to ~8.5 °C after 00:00 UTC on 14 February (Fig. 7a). The expected temperature difference between the two sites is about 6.9 °C (adiabatic cooling rate for 0.7 km height difference) when the adiabatic heating assumed. The sea level pressure also decreased from ~−1 hPa to −4 hPa at the time when the temperature was raising. The observed wind speed at the GWW site showed no obvious changes in the beginning. However, the wind speed has significantly increased just ~1 hour after the sea level pressure (temperature) was decreasing (raising). This result revealed that the changes of wind speed are possibly related to the fluctuation of temperature and pressure. To clarify the role of pressure gradient on the wind speed at the DGW site, the local accelerations between the two sites could be approximated based on the horizontal momentum equation expressed as

$$\underbrace{\frac{\partial u}{\partial t}}_{A} = \underbrace{-u\frac{\partial u}{\partial x}}_{B} \underbrace{-\frac{1}{\rho}\frac{\partial P}{\partial x}}_{C} + \underbrace{fv}_{D} + \underbrace{\frac{C_d W_s u}{H}}_{E}. \tag{10}$$

In equation (10), Term A is the changes of $u$ component with the time, which also corresponds to the wind accelerations along the west-east direction and Term B is the advective





acceleration amount to the distance ($x$) between these two selected sites. Only the $u$ component
was considered in this study since the $v$ and $w$ components could be neglected because the
environmental winds mostly comprised of westerlies (Yu et al., 2020). The PGF was indicated
by Term C, where the $\rho$ is air density and the $P$ is sea level pressure. Coriolis acceleration and
friction were indicated by Term D and Term E, respectively, where the $C_d$, $W_s$, and $H$ in Term
E are the drag coefficient, wind speed and boundary layer height. The value of drag coefficient
would most likely be a unitless constant based on Stull (1988) and was set as ~$3.9 \times 10^{-3}$ in this
study. The height of $H$ used in this study was ~150 (1500) m MSL according to the GWW
(DGW) sounding observations at 00:00 UTC on 14 February 2018 (not shown).
Basically, the wind acceleration (i.e., Term A) that is derived from the equation (10) by
adding terms from B to E is in good agreement with the fluctuation of wind speed at GWW site
(Fig. 7b). A relatively weak wind speed occurred in the beginning and is coincident with negative
and weak accelerations. Consequently, the wind speed jumped at GWW site in the speed-up stage
(i.e., shading area in Fig. 7) associated with the increased and positive accelerations (i.e., Term
A). Furthermore, the contribution of Terms B~E into Term A could also be evaluated individually
by calculating each Term. PGF (Term C) dominated the changes of Term A with almost the same
magnitudes during the entire research period as shown in Fig. 7c. In the beginning, the advective
acceleration (Term B) can provide a little bit of positive contribution to Term A while PGF term
is negative. However, both Term B and friction (Term E) gave negative feedback to Term A in
the speed-up stage. Coriolis acceleration (Term D) always exhibited an almost zero acceleration
to Term A due to sub-synoptic scale feature. The results suggested that the PGF would be the
main factor to dominate the changes of wind speed at the GWW site in the lee side.



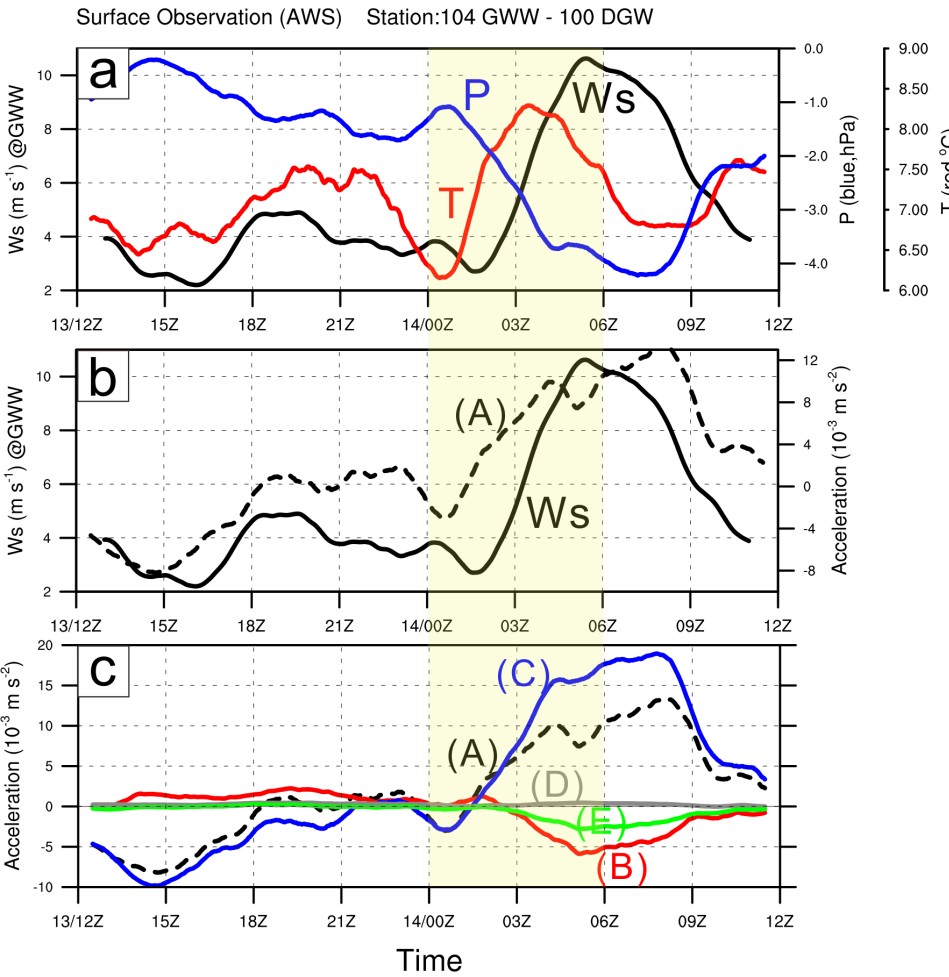


**Figure 7.** (a) Time series of wind speed (m s⁻¹, black line) observed from the AWS at GWW site, and the differences
of sea level pressure (hPa, blue line), temperature (°C, red line) between the GWW and DGW sites from 12:00 UTC
on 13 Feb. to 12:00 UTC on 14 Feb. 2018. (b) Time series of the u component acceleration ($10^{-3}$ m s⁻², Term A,
black dashed line) estimated from the horizontal momentum equation [eq. (10)] between GWW and DGW sites. (c)
Time series of the u component acceleration (Term A, black dashed line), the advective acceleration (Term B, red
line), the PGF (Term C, blue line), Coriolis acceleration (Term D, gray line), and friction (Term E, green line). The
time period with accelerating wind speed is also highlighted by light yellow shading (i.e., speed-up stage).

Since the gusty wind can be explained by PGF, this result is also consistent with the
fluctuations in the perturbation pressure from the AWS observations at GWW site (cf. Fig. 6b)
as the observed perturbation pressure always comprised of negative values with the maximum of
~ -6 hPa during the speed-up stage. To understand the possible causes of the relatively lower





pressure occurring in the lee side area, more detailed analysis is needed. The surface perturbation
pressure can be estimated by solving the integration of the vertical momentum equation (Yu and
Tsai, 2010),
$$P_S' = \underbrace{\int_0^{Z_T} \rho_0 \left( \frac{\partial w}{\partial t} + u\frac{\partial w}{\partial x} + v\frac{\partial w}{\partial y} + w\frac{\partial w}{\partial z} \right)}_{A} + \underbrace{\int_0^{Z_{LCL}} -\rho_0 g \frac{T_v'}{T_{v0}} dZ}_{B}$$

$$+ \underbrace{\int_{Z_{LCL}}^{Z_T} -\rho_0 g \frac{T_v'}{T_{v0}} dZ}_{C} + \underbrace{\int_0^{Z_T} \rho_0 g q_r dZ}_{D}, \qquad (11)$$

where $P_S'$ is surface perturbation pressure, $\rho_0$ is the reference air density function of height, and
$T_{v0}$ and $T_v'$ are the reference and perturbation virtual temperature obtained from surface
observations. The $g$ and $q_r$ in equation (11) correspond to the gravity acceleration and
rainwater mixing ratio, respectively. $Z_{LCL}$ and $Z_T$ indicate the height of the lifting condensation
level (LCL) and the cloud top, respectively. The LCL (~1900 m) can be calculated from the
averaged sounding observations at GWW during the period. The terms on the right-hand side in
equation (11) are different contributions to the estimated surface pressure ($P_S'$) including
nonhydrostatic effect (Term A), sub-cloud and in-cloud warming/cooling (Terms B and C), and
water loading (Term D).

The Terms C and D would be possibly neglected since the studied case just occurred under

fine weather condition. Remaining contributions for the estimated perturbation pressure may
come from the nonhydrostatic effect (Term A) and in-cloud warming/cooling (Term B) because
this study ignored the in-cloud warming/cooling and water loading. The estimated values of Term
B have ~1 (~2) hPa underestimated (overestimated) from the observed perturbations pressure at
~15:00 (21:00) UTC on 13 February in advance of the speed-up stage and there is about 2 hPa ~
3 hPa difference after 09:00 UTC on 14 February. Nevertheless, the estimated values of Term B
are in good agreement with the observed perturbations pressure without significant differences
(Fig. 8) in speed-up stage. The results suggested that the origin of the lower pressure in the lee
side of TMR was deduced by sub-cloud warming. Such PGF could accelerate the wind speed in
the speed-up stage in the lee side of the mountains.

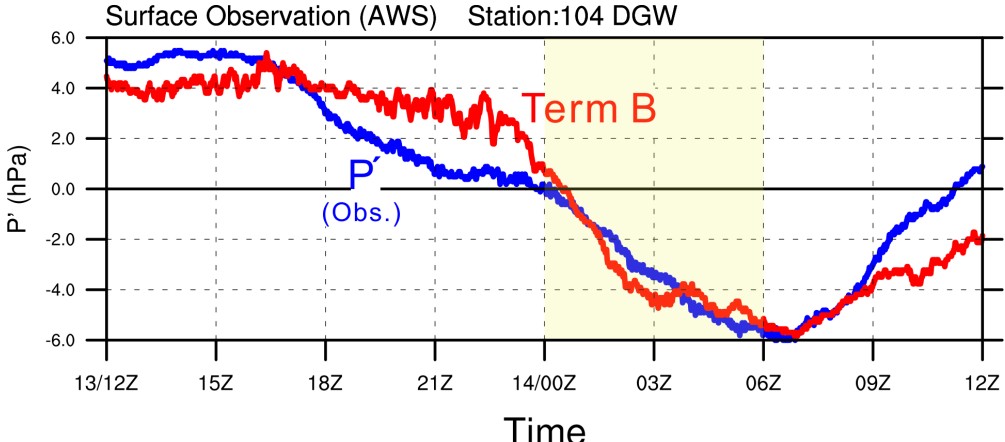


**Figure 8.** Time series of perturbation pressure (unit: hPa) observed by the AWS (blue line) and estimated Term B
(sub-cloud cooling/warming, red line) in the vertical momentum equation [eq. (11)] at the GWW site from 12:00
UTC on 13 Feb. to 12:00 UTC on 14 Feb. 2018. The time period with accelerating wind speed is also highlighted
by light yellow shading (i.e., speed-up stage).

### 4.3 Mountain wave, hydraulic jump, and downslope windstorm

The mountain wave feature was detected in the local reanalysis data of the LDAPS (Fig. 9).

Alternative downdraft and updraft were presented near the crest (near DGW site) and the lee side
of the TMR at 21:00 UTC on 13 February 2018 (at 3 hours prior to the speed-up stage, Fig. 9a).
The mountain wave propagated toward the northeastern direction (parallel to The TMR)
associated with the interactions between the prevailing west-southwesterly winds and topography
(lee wave in Fig. 9a). Stronger downdraft and updraft were characterized by positive and negative
phases stronger than 3 m s$^{-1}$ at DGW, BKC and GWW sites and the phase lines were parallel
with the orientation of the TMR. Subsequently, in the speed-up stage, the mountain wave
structures significantly changed at 03:00 UTC on 14 February 2018. The wavelength became
longer but the wave still parallel to the TMR and the northeastern coast (Fig. 9b).

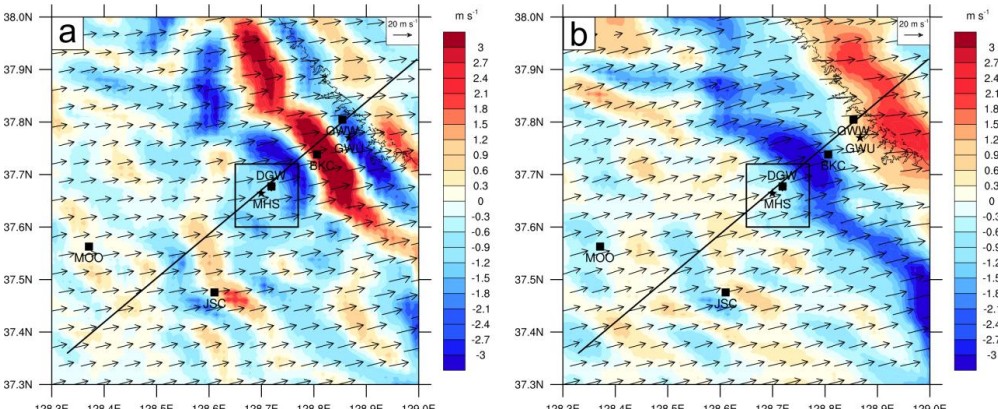


**Figure 9.** Horizontal distribution of the vertical velocity (m s$^{-1}$, color shading) and horizontal winds (vectors) at 2
km MSL from the LDAPS in the domain corresponding to the Fig. 1a at (a) 21:00 UTC on 13 Feb. 2018 and (b)
03:00 UTC on 14 Feb. 2018. The locations of the scanning Doppler lidars (soundings) sites are denoted by asterisks
(squares).

The cross section of the potential temperature (thick solid line in Fig. 10) and streamwise
velocity (colors in Fig. 10) perpendicular with the orientation of the TMR demonstrated the
mountain wave characteristics in the lee side (between the DGW and GWW sites) at 21:00 UTC
on 13 February 2018 (Fig. 10a). In this time period, a relatively stronger streamwise velocity only
occurred near the downslope of the TMR (~128.78ºE, ~1 km MSL) and coincides with the
stronger downdraft. Weaker streamwise velocity (<4 m s$^{-1}$) appeared near the GWW site in the
coastal area. However, the potential temperature pattern in the speed-up stage was characterized
by a longer wavelength with higher amplitude of the mountain wave (Fig. 10b), which is
consistent with the vertical velocity field (Fig. 9b). A stronger wind exceeding 30 m s$^{-1}$ (shaded
orange colors in Fig. 10b) stemmed from higher altitudes as jet stream was approaching the
Korean peninsula at this time. The strong wind propagated down to 0.5 km toward the GWW
site. Note that the maximum values of Froude number related to the environmental winds at the
DGW sounding site were estimated around 0.55–0.89. These Froude number were calculated
from the sensitivity test using dry and saturated Brunt-Vaisala frequency and increasing the
topographic height between 1000 and 2000 MSL. These structures were similar to the hydraulic



jump reported from several previously numerical studies in the northeastern coast of Korea (Kim
and Cheong, 2006; Jang and Chun, 2008; Lee and In, 2009; Lee et al., 2020). The stronger
streamwise velocity extended from the upper layers to the downslope of the TMR (exceeding
~36 m s$^{-1}$) coincident with downdraft, which is similar to the downslope windstorm that
contributed to the increased temperature via adiabatic warming on the slope. Along with this,
surface velocity was intensified exceeding ~12 m s$^{-1}$ near the surface at the GWW site associated
with the downslope wind. Note that the magnitudes of streamwise velocity are consistent with
the fluctuation of the surface wind speed observed from the AWS (wind speed in Fig. 6). The
result from this LDAPS analysis suggested that the downslope wind dominated the acceleration
of wind speed in the lee side of the TMR.

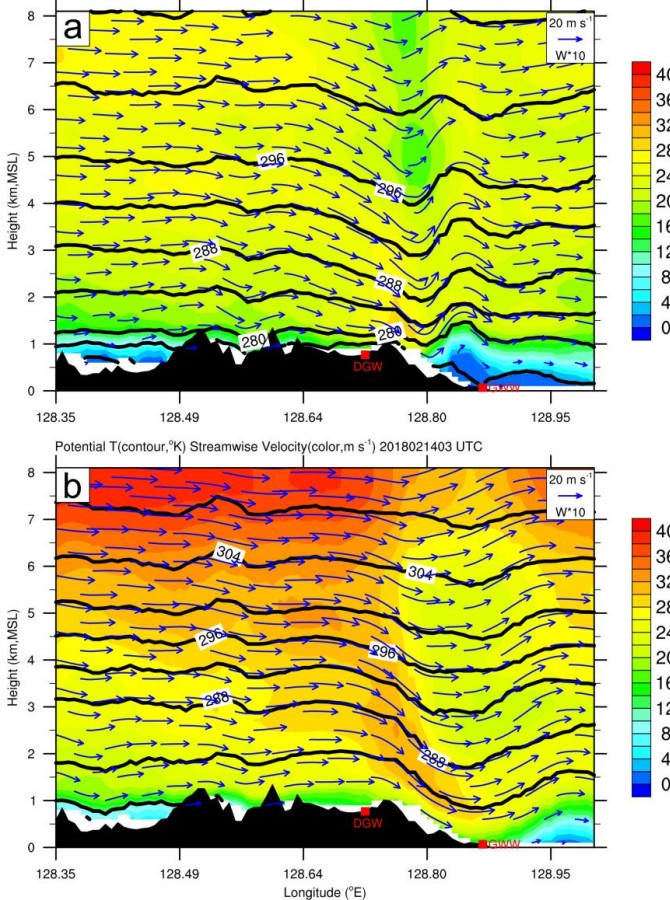






**Figure 10.** Vertical cross section of the LDAPS potential temperature (°K, contours), streamwise velocity (m s$^{-1}$,
color shading), and airflow (vectors) along the black lines in Fig. 9a at (a) 21:00 UTC on 13 Feb. 2018 and (b) 03:00
UTC on 14 Feb. 2018. The black shading in the lower portion indicates the topography along the black line in Fig.

9.

**5 Stronger winds in mountainous area**
**5.1 The lidar and AWS observations**

The combination of the LPS and HPS provided a large-scale environmental wind favorable

for westerly over mountainous area. According to the DGW QVP from observations (Fig. 11a),
the wind speed ranged from ~12 to 36 m s$^{-1}$ at the low-level layers (~900 to 1800 m MSL) during
12:00 UTC on 13 February to 12:00 UTC on 14 February 2018. After this time period, the wind
was decaying so quickly becoming nearly calm associated with the approaching HPS (Fig. 2e).
The surface wind fluctuates in the range of 7 m s$^{-1}$ to 12 m s$^{-1}$ with the periodicity of 6h at the
DGW site, similar to the pattern in the lidar QVP (Fig. 11b). These characteristics were quite
different from the AWS and lidar observations in the lee side of mountains (for example GWW
site). Unlike the coastal site, the strong wind can sustain for a day in the mountainous area. In
particular, there was persistent westerly at the entire altitude in the mountainous area (i.e., DGW
site). However, the wind direction was quite variable from southerly to westerly at the lee side of
the mountainous area (GWU or GWW site). Significant strong winds were measured at the DGW
site above 1000 m MSL on 13 February (Fig. 11), wind was weak at the GWU site (Fig. 6).
Although the wind was getting stronger at the GWU and GWW sites on ~02:00 UTC on 14
February, the low-level or surface winds are getting slightly weaker at the DGW site. This trend
is consistent with the wind fields from the sounding observations at the DGW site (Figs. 4a and
4b).

The AWS observations at the DGW site demonstrated sustained strong westerlies (~10 m s$^{-1}$)





with periodic fluctuation from 12:00 UTC on 13 February to 12:00 UTC on 14 February 2018
(Fig. 11b). Although the wind speed fluctuated periodically, no periodicity was shown in the
perturbation pressure and perturbation temperature. Instead, the perturbation pressure
monotonically dropped from 3 hPa at 12:00 UTC on 13 February to -2 hPa at 05:00 UTC on 14
February and then increased to 1 hPa at 12:00 UTC on 14 February 2018. The perturbation
temperature showed nearly opposite trend of the perturbation pressure. The perturbation
temperature was nearly steady until 22:00 UTC and then increased to 4.5 ºC at 07:00UTC on 14
February 2018. Although the movement of LPS affected the changes of the surface perturbation
pressure and temperature at the DGW site, the changes of the wind speed had no clear relation
with the surface pressure and temperature.

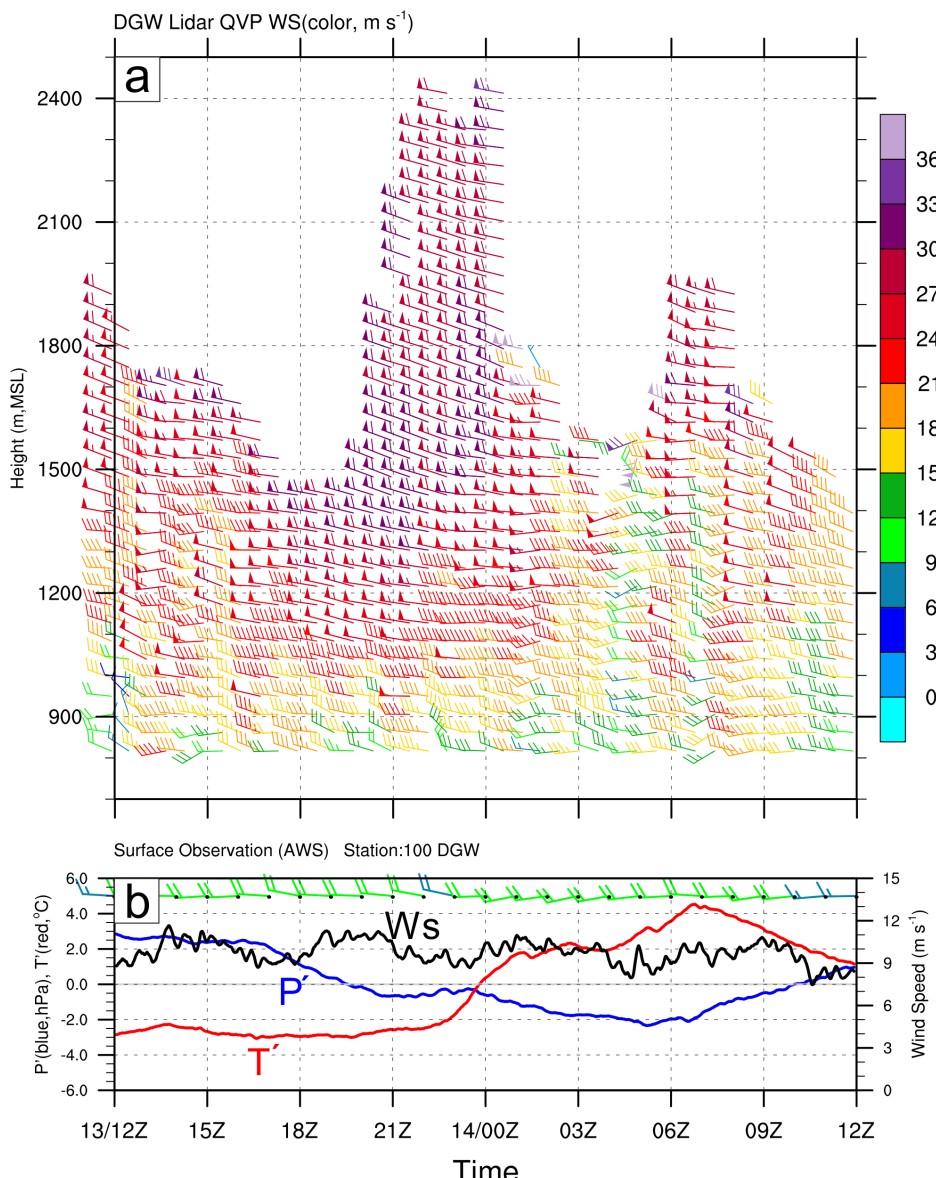

**Figure 11.** (a) Time series of QVP (Quasi-Vertical profile) from lidar observations at DGW site from 12:00 UTC on 13 Feb. to 12:00 UTC on 14 Feb. 2018. A full wind barb corresponds to 5 m s$^{-1}$; a half barb corresponds to 2.5 m s$^{-1}$ and the color indicates the wind speed correspond to bar. (b) Time series of the horizontal winds (wind barbs), wind speed (m s$^{-1}$, black line), perturbation pressure (hPa, P', blue line), and perturbation temperature (°C, T', red line) observed from the AWS at DGW site.





**5.2 Possible mechanism of strong wind in mountainous area**

To document the possible mechanisms about sustained strong wind occurring at DGW site over the mountainous area, difference of temperature and pressure were analyzed in detail. Similar to the DGW and GWW sites in Fig. 7, an upstream surface station (YPO site in Fig. 1b) was selected to calculate the temperature and pressure differences with the DGW site. Figure 12a reveals that the fluctuation of pressure differences (blue line in Fig. 12a) had an almost out of phase with the fluctuations in wind speed (black line in Fig. 12a) at the DGW site. Furthermore, the wind speed was gently decreasing with periodicity (wavelength of about 6h).This result provided a clue that the pressure gradient likely dominated the wind speed in this local area. Compared to the lee side of the mountains at the GWW site (Fig. 7), negative values of the temperature differences (minimum −1.3 ℃) were calculated in the mountainous area and even became smaller (−0.5 ℃) after 12:00 UTC on 14 February. Thus, the differences of pressure seemed to affect the wind speed patterns, and the fluctuation of wind speed did less relate to differences of temperature between these two sites. Note that the estimated values of Team B in eq.(11) are revealed significant different from observed perturbation pressure at the DGW site (between ~1 hPa and 2 hPa) during entire period, this result also indicated that the fluctuation of pressure is more related to nonhydrostatic effects [Term A in eq. (11)] but sub-cloud warming/cooling (not show). The periodic characteristic of the surface wind may have linked to nonlinear dynamic such as gap flow and gravity wave mechanisms (Shun et al., 2003).

The acceleration of wind speed at the DGW site can also be estimated by equation (10). Most of the estimated Term A and wind speed were also in a good agreement except for the short time period (Fig. 12b). Basically, the wind speed was increasing (decreasing) when estimated acceleration (Term A) is positive (negative). To understand the main contributor that dominates such local strong winds in this area, a detailed budget analysis of the momentum equation was performed (Fig. 12c). The PGF (Term C) was the most important factor for the estimated



acceleration, which means that the PGF could possibly determine the changes of the wind speed
at the DGW site. The advective acceleration was relatively small. The Coriolis force and friction
had no clear impacts on the acceleration (Term A).

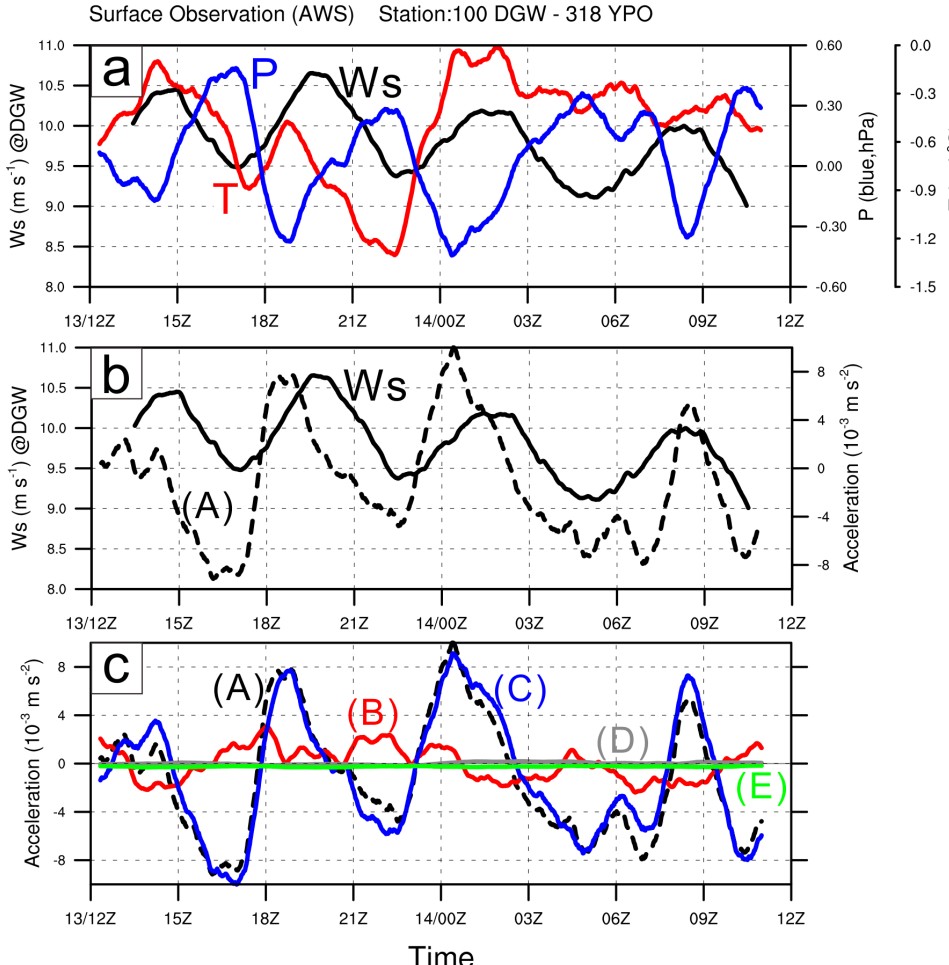


**Figure 12.** (a) Time series of wind speed (m s$^{-1}$, black line) observed from the AWS at DGW site, and the differences
of sea level pressure (hPa, blue line), temperature (ºC, red line) between the DGW and YPO sites from 12:00 UTC
on 13 Feb. to 12:00 UTC on 14 Feb. 2018. (b) Time series of the u component acceleration (10$^{-3}$ m s$^{-2}$, Term A,
black dashed line) estimated from the horizontal momentum equation [eq. (10)] between YPO and DGW sites. (c)
Time series of the u component acceleration (Term A, black dashed line), the advective acceleration (Term B, red
line), the PGF (Term C, blue line), Coriolis acceleration (Term D, gray line), and friction (Term E, green line).

The above results show that PGF is the main factor to accelerate wind speed, but temperature

is not a critical factor to change the PGF over the mountainous area. To determine the possible





factor that contributes into PGF, more detailed analysis of horizontal winds was performed with
WISSDOM synthesis. Figure 13 demonstrates the fine-scale wind fields at 800 m MSL (near
surface in the studied domain). In this height, a unique topographic feature can be explored as it
is composed of a relatively wide (narrow) area in the western (eastern) side along the valley. This
channel like feature could be emphasized and was marked by the area between two thin dashed
lines in Fig. 13. Four periods (00:00 UTC on 13 February 2018, 00:00 UTC on 14 February,
12:00 UTC on 14 February, and 00:00 UTC on 15 February 2018) were selected to investigate
the changes of wind patterns in this channel along the valley. The prevailing wind was westerly
with a slight deflection near the center of the domain and the eastern side of the valley while the
LPS approached Korea (Figs 13a, 13b, and 13c). Nevertheless, a relatively weak wind (~6 m s$^{-1}$)
always existed in the center of the domain nearby the MHS lidar site (wide segment of the valley)
and a stronger wind (14 m s$^{-1}$) was observed in the downstream nearby the DGW site (narrow
segment of the valley). Wind speed decreased and nearly became calm after the LPS moved away
from Korea (Fig. 13d).

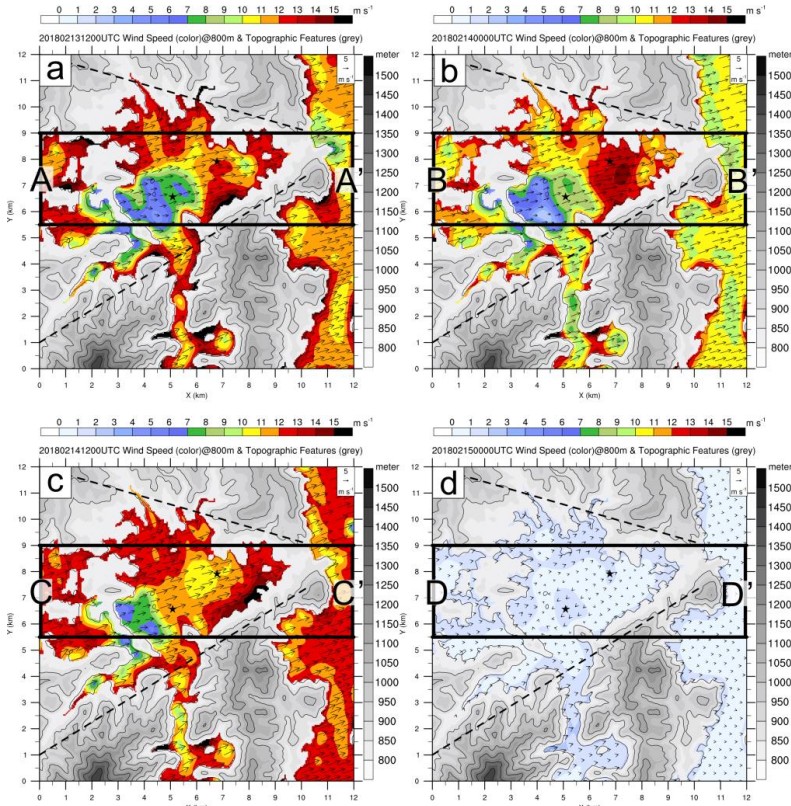

**Figure 13.** Horizontal distribution of the wind speed (m s⁻¹, color shading) at 800 m (MSL) retrieved in the WISSDOM domain at (a) 00:00 UTC on 13 Feb. 2018, (b), 00:00 UTC on 14 Feb., (c) 12:00 UTC on 14 Feb., and (d) 00:00 UTC on 15 Feb. 2018. The black dashed lines mark the area of the channel to calculate the averaged wind speed and channel width as shown in Fig. 14. The inserted box indicates the averaged area in the vertical cross sections along the valley (A-A'). Topographic features indicated by the gray shading and contours. Locations of the scanning Doppler lidar sites are denoted by asterisks.

To understand the relations between the topography and winds, the average wind speed (colored thick lines in Fig. 14) and the channel width (thick black line in Fig. 14) along the valley at 800 m MSL were calculated in two time periods when the LPS approaches (before 12:00 UTC on 14 February 2018) and leaves (after 12:00 UTC on 14 February 2018). The channel width was around 2 km at x = 0 km to 3 km (western side) and became wider ~5.5 km at x = 3 km to 6.5 km. The channel width then decreased significantly to nearly 0 km at x = 6.5 km to 9.5 km.

When LPS was approaching (average wind speed in red line and range of minimum and



maximum wind speed in shading in Fig. 14), the average wind speed increased from ~10 m s$^{-1}$
to ~14 m s$^{-1}$, which was coincident with the changing of channel width from ~5.5 km to 0 km
along the valley. The maximum wind speed was larger than 24 m s$^{-1}$ near the narrowest segment
of the valley. When LPS was leaving (blue line and shading), the averaged wind speed had a
nearly constant strength (~5 m s$^{-1}$) without obvious changes and no clear relations with the width
of the valley. This analysis reveals that the channeling effect may play an important role to
dominate the spatial distribution of wind speed with the valley. Furthermore, downstream
acceleration in the narrow segment of valley could be more significant with strong upstream
winds.

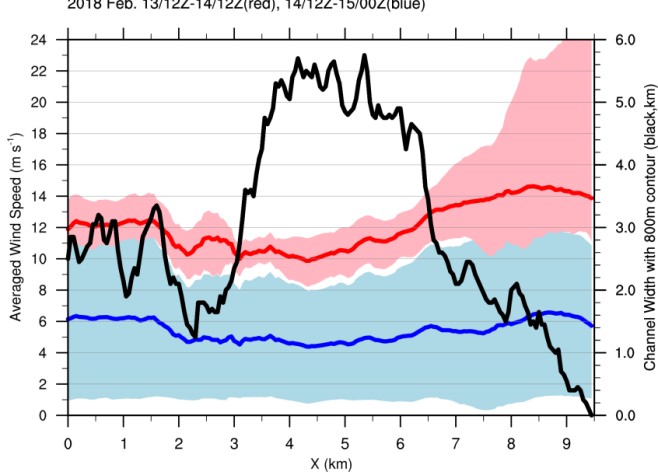


**Figure 14.** Averaged wind speed and its range along the valley corresponding to the area indicated by the dashed
lines in Fig. 13 at two times: 12:00 UTC on 13 Feb. to 12:00 UTC on 14 Feb. (red line and shading) and 12:00 UTC
on 14 Feb. to 00:00 UTC on 15 Feb. 2018 (blue line and shading). The red and blue shading show the maximum and
minimum values along the valley for the two times. Averaged channel width along the valley was plotted by a thick
black line.
Figure 15 shows the mean vertical structures of wind speed, airflow, and topographic features
from each cross section along the boxes in Fig. 13. The boxes were set on our main focus area
from wider to narrow segments along the valley and parallel with the environmental wind
direction (westerly). These analyses allow us to investigate detailed airflow features from near
the surface to higher altitudes and their interactions with topography. The four time periods were
12:00 UTC on 13 February 2018, 00:00 UTC and 12:00 UTC on 14 February, and 00:00 UTC
on 15 February 2018. The mean vertical structures in the first three periods (when the LPS
approaching) revealed similar characteristics that the uniform and stronger westerly winds (larger
than ~18 m s$^{-1}$) were above the layers higher than 1 km MSL. In contrast, the airflow had more
significant variances near the surface layers. In the layers below 1 km MSL, the westerly winds
were lifted on the upslope and become downdraft behind the mountain crests. The wind speed
was quite weak (strong) near the MHS (DGW) site, which is coincident with the relatively wider
(narrow) segment in the valley on the three time periods. In particular, the high wind speed area
was only presented between x = ~6.5 km and 9.5 km (i.e., the narrowest segment of the valley).
The winds became more uniform and weaker in the upper layers and near the surface when the
LPS moved away from Korea at 00:00 UTC on 15 February 2018 (Fig. 15d).

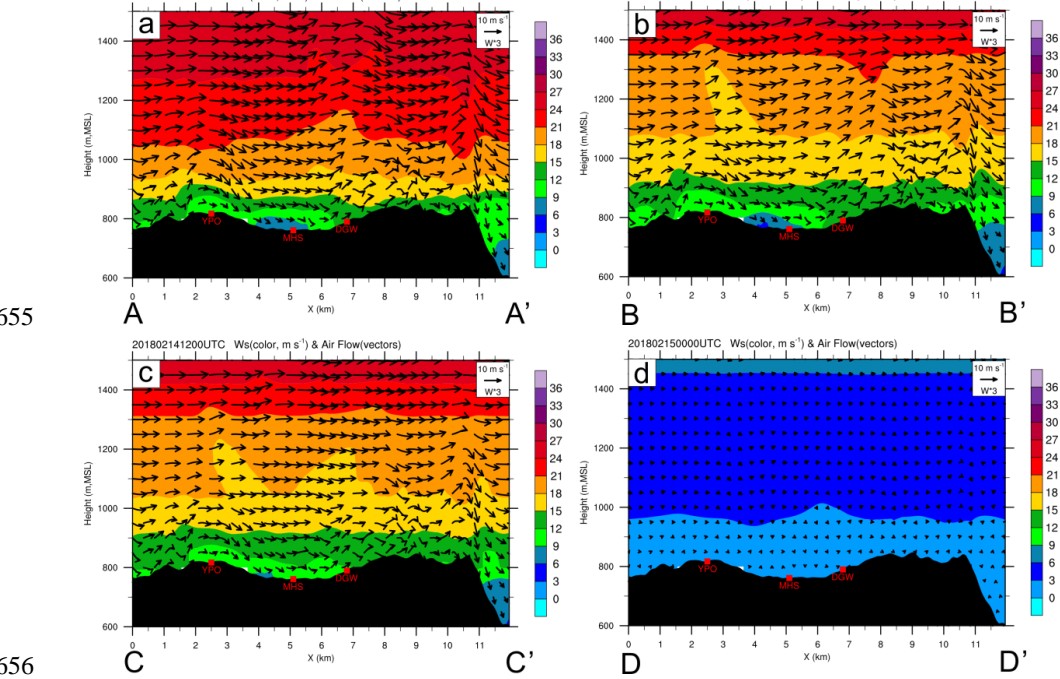



**Figure 15.** Averaged vertical cross section of the WISSDOM-derived wind speed (m s$^{-1}$, color shading) and wind
vectors (combined cross barrier flow and threefold vertical velocity) at four time periods (a) 12:00 UTC on 13 Feb.
2018, (b) 00:00 UTC on 14 Feb. 2018, (c) 12:00 UTC on 14 Feb. 2018, and (d) 00:00 UTC on 15 Feb. 2018. The





area of the cross section is shown in the black box in Fig. 13. The black shading in the lower portion indicates the
topography along the box.
Because the winds manifested clear variations only near the surface layers, the mean vertical
structures of wind speed and directions could be further averaged just below 1 km MSL. Fig. 16
shows the continuous time series of averaged wind field during the entire period with the same x
axis in Fig. 15. Results demonstrate that the winds near the surface layers were accelerated in the
narrow segment between x = ~6.5 km and 9.5 km for strong enough upstream winds (before
00:00 UTC on 14 February). This characteristic is similar to the gap wind or channeling effect
from the previous simulation and observational studies (Overland and Walter, 1981; Neiman et
al., 2006; Heinemann, 2018). Consequently, a relatively weak channeling effect induced weaker
winds in the narrow segment of valley during 00:00–15:00 UTC on 14 February 2018 because
the environmental winds from upstream became weaker. Finally, the channeling effect was no
longer exiting when the upstream winds became calm after 15:00 UTC. The wind would possibly
accelerate when it blows from wider to narrow segments of the valley due to PGF as it is related
to the Bernoulli's Law, i.e., the pressure would reduce when the flow speed is increased and vice
versa. Observational analysis reveals a relatively low pressure in the narrow segment of valley
and thus, PGF would locally dominate the airflow accelerating over the mountainous area.



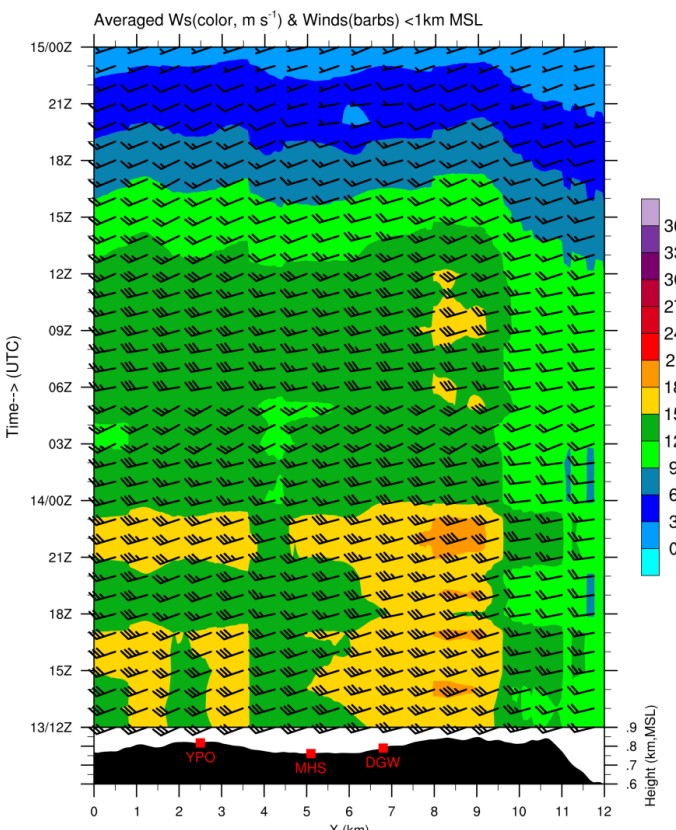

**Figure 16.** Temporal variation of the averaged wind speed (m s$^{-1}$, color shading) and the horizontal winds (wind barbs) from WISSDOM derived in the valley from 00:00 UTC 13 Feb. on 13 to 00:00 UTC on 15 Feb. 2018. The low-level winds (below 1 km MSL) within the black boxes in Fig. 15 were averaged in a direction normal to the orientation of the boxes. The black shading in the lower portion indicates the averaged topography along the boxes.

## 6. Conclusion

This study uses Doppler lidars, wind profiler, soundings, and surface observations to examine a strong downslope wind event during the ICE-POP 2018. Detailed characteristics of wind fields and possible mechanisms were explored during the passage of a low-pressure system (LPS) over the northern part of the Korea Peninsula on 13–15 February 2018. Although the wind speed is generally increased in South Korea when the LPS is approaching, it comprised of more significant increasing trend along the downslope and in the lee side of the TMR (Taebeak



689 Mountain Range). The wind speed has no obvious changes but are persistently strong over the

690 TMR. Conspicuous gradient of linear trend of the wind speed only existed between the

691 mountainous areas and in the lee side areas. Moreover, the wind speed shows a decreasing trend

692 synchronously after the LPS moves away from Korea.

693  From the sounding observations, low-level environmental winds revealed high variability

694 from the mountainous area to the lee side of the mountains. The wind direction is comprised of

695 most westerly associated with the LPS and the wind speed are sustainedly strong (~10 m s$^{-1}$) at

696 the DGW site (i.e., mountainous area) during the research period. However, the wind speed at

697 the lee side (GWW) clearly changed from being relatively weak to a stronger one. The winds

698 then become nearly calm both in the mountainous or lee side areas after the LPS moved away

699 from the Korean peninsula. In addition, upstream inversion layers (at ~850 hPa level) were also

700 detected by sounding observations at the DGW site while the strong wind occurred in the lee side

701 of the mountains.

702  In the lee side of the mountains, the surface wind speed has dramatically increased (from ~3

703 to 12 m s$^{-1}$) at the GWW site during the research period. The surface temperature (pressure)

704 perturbation is also changed to positive (negative) values and showed significant time lag with

705 wind speed change. The sea level pressure and temperature differences between the mountainous

706 station at the DGW and the lee side station at the GWW demonstrate that the wind speed is

707 suddenly raised with increasing temperature (exceeded ~8.5ºC) and decreasing pressure (from −1

708 hPa to −4 hPa). The estimated wind accelerations [Term A in eq. (10)] are in good agreement

709 with the observed wind speed, which are mainly contributed by the PGF [Term C in eq. (10)].

710 The negative surface perturbation pressure at the GWW site has nearly the same magnitude with

711 the estimated surface perturbation pressure from the integration of the vertical momentum

712 equation. The fluctuation of surface perturbation pressure is mainly dominated by the sub-cloud

713 warming/cooling [i.e., Term B in eq. (11)]. Results indicate that the adiabatic warming plays an

714 important role to reduce the surface pressure and the winds are accelerated by PGF in the lee side



of the TMR. Furthermore, the downslope winds were also dominated by the stronger wind
occurring along the lee side based on the LDAPS analysis. The development of the strong
downslope wind is highly related to the mountain wave and hydraulic jump.

In the mountainous area, persistent surface strong wind (with the fluctuation of 4.5 m s$^{-1}$)

was observed at the DGW site when the LPS was approaching (leaving). The surface wind has
no clear relationship with the surface perturbation pressure and perturbation temperature.
However, the sea level pressure differences between the upstream station YPO and DGW show
similar amplitudes (out of phase) in the fluctuation of surface wind speed. In contrast, the
temperature differences are small (between −0.5°C and 1.2 °C) with no clear relations with the
fluctuation of surface wind at the DGW site. Although the temperature has no clear relation with
the strong wind, estimated wind accelerations [Term A in eq. (10)] results are in good agreement
with the observed surface wind speed. It means that the PGF is still the main contributor for the
wind acceleration at the DGW site. The 3D winds derived from WISSDOM synthesis also reveal
that the wind speed at the DGW site (narrow segment in the valley) is always stronger than the
YPO site (wider segment in the valley) when the LPS was approaching. Thus, the channeling
effect is possible mechanism to dominate the wind acceleration in the mountainous area.

In this study, the observational evidence shows that the different mechanisms are important

references to determine the strength and persistence of the orographically strong winds in the
same underlaying LPS under fine weather condition. In the future, high-resolution numerical
modeling analysis will be performed for all strong wind events during the ICE-POP 2018,
because the detailed thermodynamic information was desired to give more complete descriptions
about the distribution of potential temperature across the mountainous area. The kinematic and
thermodynamic information from the simulations will be important indicators to further
investigate the existing of mountain wave, included the hydraulic jump, wave breaking, and
partial reflection for the generation of the downslope windstorm. More cases will be included to
provide comprehensive explanations of the strong downslope wind in the northeastern



mountainous part of South Korea. More importantly, we aim to extend our understanding on the
variability of winds around the terrain in a very fine-scale even at different seasons.

*Author contributions.* This work was made possible by contribution from all authors.
Conceptualization, CLT, GWL, JHK ; methodology, CLT, YCL, YHL, JHK, and KK; software,
CLT, YHL, and KK; validation, KK, YHL, and GWL; formal analysis, CLT, and JHK;
investigation, CLT, GWL, and YHL; writing—original draft preparation, CLT; writing—review
and editing, GWL, JHK, YCL and YHL; visualization, CLT; supervision, GWL, and YHL;
funding acquisition, GWL, YHL, and JHK. All authors have read and agreed to the published
version of the manuscript.

*Competing interests.* The authors declare that they have no conflict of interest.

*Special issue statement.* This article is part of the special issue "Winter weather research in
complex terrain during ICE-POP 2018 (International Collaborative Experiments for
PyeongChang 2018 Olympic and Paralympic winter games) (ACP/AMT/GMD inter-journal SI)".
It is not associated with a conference.

*Acknowledgments.* This work was supported by Civil-Military Technology Cooperation Program
funded by the Korea Meteorological Administration and Defense Acquisition Program
Administration. No. 17-CM-SS-23, KMA2017-04210, [Project Name : Development of fusion
technology for Radar wind profiler] and was funded by the Korea Meteorological Administration
Research and Development Program under Grant KMI2020-00910. The authors greatly
appreciate the participants in the World Weather Research Programme Research Development
Project and Forecast Demonstration Project, International Collaborative Experiments for
Pyeongchang 2018 Olympic and Paralympic winter games (ICE-POP 2018) hosted by Korea
Meteorological Administration (KMA). The Doppler lidars were deployed by the National
Institute of Meteorological Sciences (NIMS), KMA and Environment Climate Change Canada
(ECCC). We would like to thank many researchers and students (Byung-Chul Choi, Kwang-Deuk
Ahn, Namwon Kim, and Seung-bo Choi at KMA, and Choeng-lyong Lee, Daejin Yeom, Kyuhee
Shin, DaeHyung Lee, Su-jeong Cho, SeungWoo Baek, Hong-Mok Park, Geunsu Lyu, Eunbi
Jeong, Heesang Yoo, Youn Choi, Bo-Young Ye, and Soohyun Kwon at Kyungpook National
University) who collected data during the ICE-POP 2018 period.



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
