# Peer review of "Observational study for strong downslope wind event under fine weather"

_Atmospheric Chemistry and Physics, 2021_

## Author Comment (AC1)

**acp-2021-100**
**Responses (highlighted with blue) to Referee #1**
**31 May 2021**

Review of manuscript entitled "Observational study for strong downslope wind event under fine weather conditions during ICE-POP 2018" from Tsai et al.

**Recommendation**: Reconsider after major revisions

**Summary**

The manuscript discusses observations obtained during a downslope wind event in a coastal mountainous setting, the Taebeak Mountain Range in eastern South-Korea, during the winter of 2018. The manuscript aims to explain the acceleration of winds in the lee slopes in a coastal setting, using data obtained in an upstream environment that encompasses a valley that narrows towards the coast. While the data seems quite abundant, and the authors have clearly done an extensive job in figure creation and additional analysis, it is unclear what scientific problem the manuscript aims to discuss. There are also quite a few unclear steps taken in the analysis approach, which need to be addressed. I'm in between major revisions or reject, but want to give the benefit of the doubt at this stage to enable the authors to improve their manuscript substantially. Please refer to the comments below.

We appreciate Referee #1's helpful and constructive comments, which help us to improve the manuscript substantially. We have more emphasized the importance of our study and what scientific problems want to address, the capability of adopted datasets and their specific steps of data processing have been also clarified in our revised manuscript. A set of responses to the reviewer's comments is provided below and specific locations of revised portions were also noted as the number of lines.

**Major comments:**
- Data and period selection. Although the authors state that this day was chosen because some parts of the Olympic games were postponed, it would be interesting to know how this relates to climatology of wind events in the area. That would emphasize better the importance of the study. Was this the strongest wind event in the lee slopes at WWG? Or was it just the only event that could be considered as strong?

This is a particularly good comment. Except for the Olympic games were postponed, our selected event is one of two extreme wind events in the past decade based on the KMA (Korea Meteorological Administration) observational records. Besides, the persistent strong wind occurred frequently in narrow segment along the valley (i.e., near the DGW site) from the record. The climatological information also manifested the importance of the large-scale weather systems in this extreme wind event. Additionally, snice the dense observational network was built during ICE-POP 2018, this is a good opportunity to investigate this unique event. We have emphasized these in the introduction (Section 1, Lines 139-142) and provided detailed descriptions about the KMA record in last paragraph of Section 3.1 (Lines 352-367).

- The study presents a mixture of model simulations and observations, but this is not clear from title, abstract or methodology section, and should be emphasized. More important in this comment is that at times it is unclear whether the authors present observations, simulations or both? In the end, once the WISSDOM is used, this is a mix of observations and numerical output and therefore the study cannot be presented as observations alone. Additionally, not much is discussed regarding the WISSDOM data (how accurate is the approach?), nor the inclusion of the numerical model data into the WISSDOM. Science is about understanding the uncertainties in the data presented, but the authors do not seem to discuss any of it.

Various datasets were acquired in this study including conventional observations [scanning Doppler lidar, automatic weather stations (AWss) and soundings] and reanalysis data (ERA5 and LDAPS). In fact, the forecast outputs of the LDAPS does not adopted in our analysis and it does not to be the constraint in WISSDOM as well. This study is aimed to examine the evolution and mechanisms of an extreme strong wind event associated with a passing LPS (low-pressure system) in Korea with abundant data and reanalysis data. From this statement, the title was modified to "**An analysis of an extreme wind event in a clear air condition associated with a low-pressure system during ICE-POP 2018**" for clarity (Lines 1-2), and we have also emphasized this in abstract (Lines 37-41), introduction (Lines 133-137), and provided more detailed information about the ERA5 and LDAPS datasets in Sections 2.3 (Lines 220-241).
Additionally, we explained the role of the LDAPS in WISSDOM in this revision. Its constraint is used to minimize the squared errors between the horizontal winds of LDPAS and synthesis winds of WISSDOM. Thus, the role of the LDPAS winds in WISSDOM is to improve the accuracy of the retrieved winds (the details have been noted in Lines 288-292). The accuracy of WISSDOM's winds was also discussed by previously studies, the retrieved winds reveal good relations and acceptable

discrepancies (maximum correlation coefficient is 0.86, minimum root mean square deviation is 1.13 m s$^{-1}$) compared with conventional observations (the descriptions have been also added in Lines 293-305).

- Some critical explanation of data usage and data treatment is missing. For example, a trend of wind change represented as a percentage per hour is maybe a different way than normal, but just in the sense of diurnal variability it does not make sense. Wind speeds at the surface change over the course of less than 15 hours (the time frame the authors chose for this figure), and so this could also be clearly within the diurnal variability of winds. Is it just a trend based on hourly data? Second, it is unclear what perturbation pressure and temperature represent, as these are not defined. Third, there is a nationwide plot that presents AWS stations, but these are not introduced in the data and methods section. There are some other examples that I leave to the authors to read in minor comments below.

The response to the first comment: to avoid the possible diurnal effects on the wind speed observations, we provided a better analysis to explain the changes of surface wind speed during research period. A sequence of figures (Line 368, Figure 3) shows clear evolution of surface wind speed in northeastern region of Korea and their relations with the moving LPS. The descriptions about these changes have been revised in third paragraph of Section 3.1 (Lines 339-351).
The response to the second comment: since the station pressure and temperature can better represent the of local ambient (follow a Minor comment for "Figure 6" below). Therefore, we use station pressure and temperature for further analysis instead of original one (Lines 450-467, Fig. 6b, and Lines 623-632, Fig. 11b).
The response to the third comment: data processing and the characteristics of these nationwide AWS observations were introduced in Section 2.2 (Lines 174-191). There were 727 regular and additional 32 AWS stations in Korea (mean distance is ~10 km for each station), the AWS observations have to interpolate to the given grids by objective analysis with the influences of radius in 10 km.

- A few statements made in the paper seem incorrect or speculative of nature. Please see comments below.
- The manuscript is quite poorly structured. Section 2 with 2.1 explaining lidar on itself is very dense, while 2.2 is a combination of brief explanation of AWS, sounding, wind profiler and model (!) simulations. Section 3, 4 and 5 basically contain the full results part and could possibly be combined in one section. In the present structure, it is hard to understand what problem the manuscript is trying to address.

Thanks for constructive comments. We have rearranged the structures of this manuscript to clarify the main purposes (i.e., the evolution and mechanisms of strong winds over complex terrain) of this study. Start from Section 2, the detailed introductions of conventional observations (AWS, sounding, wind profiler) and reanalysis datasets (ERA5 and LDAPS) were be separated into two different sections (Section 2.2 and 2.3). In the Section 2.3, the general bases of the LDAPS and ERA5 were clearly addressed, and their spatiotemporal resolution were also noted. The changes of AWS wind speed and their climatologic information have been revised in Section 3.1. In Section 4, we combined the contents from original one (Section 5). The descriptions about evolution of the strong winds in the leeward side of the mountain range, and the LDPAS analysis were switched to Section 4.1. The descriptions about evolution of the persistent strong winds in the upstream of the mountain range were switched to Section 4.2. The explanations for their possible mechanisms were moved to two subsections 4.1.2 and 4.2.2. Finally, the conclusion is in Section 5.

- The manuscript is full of grammatical errors, and the phrasing is hard to read. I highlighted only a few, but please let it proofread by an English native speaker, or perhaps pass it through a professional editorial company. While one can clearly not argue on writing style, the text has to be comprehensible, and, unfortunately, in the current state this is not the case.

  Thank you for pointing out this problem, we have already checked the grammatical errors carefully. The manuscript has also been edited by professional editorial company.

- There are a handful of studies that use lidar observations to explain downslope windstorm events. Please include these studies. A simple web search would suffice here.

  Thank you for this suggestion. We have included four additional studies in the introduction of this manuscript (Lines 120-122). They all utilized lidar observations to document the downslope windstorm.

- A textual note is that the paper is full of abbreviations. Please consider introducing a table that would summarize instrument platforms and locations. This would help the reader greatly to refer back to.

The names, instruments, temporal resolution, locations and altitude of adopted stations have been summarized in Table 1 (Line 211). The temporary and permanent observations were also noted in the table.

**Minor comments:**

Line 37. Fine weather… What is fine weather? Fair weather? Or just pleasant weather? In the latter, one would not expect much wind… Or is it related to cloudless skies? Please be specific.

We changed these words to "clear air" for clarity based on glossary of AMS (American Meteorological Society). Since there are only strong winds but no precipitation in our selected event, the scanning Doppler lidars and the observations allow us to collect more wind information under clear air condition. All of the words (fine weather) have been replaced by "clear air" throughout the manuscript.

Lines 60-61. It seems very strange to start a paper describing a downslope wind event with a precipitation statement. Suggest to delete this phrase.

Removed as suggestion (Line 61).

Line 67. Fine weather… What is this?

Reworded to "clear air" throughout the manuscript.

Line 78. "usually occurs at the lee side". By definition, the downslope windstorm occurs at the lee side a mountain range. Please correct.

Corrected as suggestion (Line 78).

Line 80-81. "explained by hydraulic jump". Please correct to "accompanied with hydraulic jumps".

Corrected as suggestion (Lines 81-82).

Lines 116-117. "Wider … conditions." Redundant phrase. Please delete.

Removed as suggestion (Line 119).

Line 118. "the best solution". Arbitrary statement. Please change to "one approach to obtain more complete wind data is the use of Doppler wind data".

Revised as suggestion (Lines 119-120).

Line 157-169. This is a nice overview that is somehow lacking for any of the other observational platforms.

More complete and detailed introductions for observations, reanalysis datasets and the principle of WISSODM have been improved in following sections (from Section 2.2 to 2.4).

Line 165. "100 km". This is probably not true, please address.
The redundant descriptions have been removed for clarity (Lines 166-167).

Line 169. "0.04". Why this value?
We have done with many tests from 0.01, 0.02, 0.03 ~ 0.1, and this value can appropriately remove most noises and retain sufficient meteorological signals. This explanation has been added in Lines 171-173.

Line 170. LDAPS is derived from model simulations? This needs to be emphasized, as it looks now as if this is an observational dataset.
We have improved the introductions of the LDAPS dataset detailly (as Section 2.3). The forecast outputs of LDAPS dose not used in this study, the capabilities and spatiotemporal resolution were also addressed in this section (from Line 220). Please refer to our responses in second major comment above.

Lines 171-177. Please provide more detail on measurement height and other instrument details. For some reason, these are only provided for the doppler lidar. Were the sounding stations only added for the field experiment time, or are these permanent stations? Was there missing data? Were the soundings always launched at increments of 3 hour? There must have been some discrepancy in release times, but there is no information. It would also be nice to show a table with available observational platforms that accompanies figure 1, for example.
We have provided detailed information about adopting datasets in Sections 2.2 and 2.3 (Line 174 and Line 220), and these information have been summarized in Table 1 (line 211).

Line 177. "are" is "were". Five soundings at one time at all locations? Please let someone proofread.
Revised as suggestion (Line 195), this sentence has been rewritten for clarity (Lines 194-197).

Line 180. What is an "environmental wind"? Please define "very fine-scale".
We want to explain such dense sounding observations, which can represent local horizontal winds in relatively small scale (~15 km). The descriptions and the definition of these two words have been improved in Lines 201-205.

Figure 1. Please include an inset map of South Korea to indicate where this is (figures should be standalone). Presumably the white area in (a) is the ocean? The colormap suggests this is a mountain. It would be good to have a table in addition to this figure to indicate the abbreviations and the platforms used.

The Korea map has been inserted in Figure 1 (Line 212). The color bar in Figure 1 were modified to be corrected one. The names of each station in Figure 1 were also summarized in Table 1.

Lines 192-197. LDAPS is a numerical model. It is misleading to have this included in an observational paper without really emphasizing this. The title of the paper reads "Observational study", besides the model simulations are not mentioned in the abstract. It also remains unclear whether this is based on reanalysis, or whether this is a forecasting tool. This is important, as the results are presented as an observational study, but the model at 1.5 km grid spacing will never represent the terrain in such an accurate manner that one can present these results as observations. How is the data corrected regarding the terrain smoothing in the model?

We have modified the title because the conventional and reanalysis datasets were used in this study (Lines 1-2). We have emphasized what kind of the datasets were adopted in the abstract as well (Line 37-41). The LDAPS reanalysis dataset was assimilated by various platforms with high resolution of wind observations (like lidar, AWS, sounding, wind profiler and satellite). The errors between conventional observations and LDAPS have been minimized conscientiously by the KMA and the quality of wind information is able to resolve small-scale weather phenomena over complex terrain in Korea. The detailed descriptions about the LDAPS dataset have been revised in Section 2.3 (Line 220).

Line 249. "fine weather condition". See above.
Reworded to "clear air" throughout the manuscript.

Line 251. This must be plural, please address.
Revised as suggestion (Line 307).

Line 258. Stronger than what?
This a redundant word and it has been removed (Line 308).

Figure 2. Please modify the caption such that it reflects (a,b,c, etc).
The caption of Fig. 2 has been modified (Lines 320-324).

Line 271. "Consequently."

Already checked, and it looks a correct use (Line 325)

Line 276. "The other … from China". Awkward phrasing, please address.

The sentence has been rewritten (Lines 330-333)

Line 278. "northerly winds". Is this in figure 2e and 2f? Please make a reference.

The reference has been made, and more clear description was also added (Lines 332-333).

Line 280. Not sure why this is important in "fine weather conditions". Was there precipitation elsewhere on the peninsula?

Because lidar is not like radar, it will have severe attenuations when raining or snowing. So, it is a big challenge to collect the good coverage of wind information under clear air condition. Fortunately, many observational platforms were deployed at the time when the extreme strong wind occurred during ICE-POP 2018. We have emphasized this point of view in Lines 335-338.

Lines 282-294. It is arbitrary to use a trend for wind speeds at the surface over the course of only 15 hours, this is clearly within the diurnal variability of winds. How was this calculated? Also, where does this data come from? See also comment on figure 3 below.

Instead of the trends for wind speed, we use consecutive wind speed analysis during research period to explain the relations between wind speed and the LPS. The results shows that the changes of wind speed have clear relations with large-scale weather system and reveals relatively weak relations with diurnal effects. In particularly, the wind speed was increasing when the LPS was passing and was decreasing when the LPS was moving away the Korean peninsula. This new analysis can also be sufficiently presented the uniqueness as the sustained (gusty) strong wind occurred over mountainous area (lee side of the mountain range). The new figure (Figure 3, Lines 368-372) and the detailed descriptions (Lines 339-351) about the changes of wind speed have been revised in Section 3.1.

Line 284. "leaving". Awkward, please rephrase.

This paragraph has been rewritten to clarify the changes of wind speed during research period.

Line 286. "these two stages *are* shown in Fig.". Please use present tense when you refer directly to the figure, and past tense when you describe the event that occurred in the past. There are many grammatical errors like this, please address.

We have checked the grammatical errors and have been corrected throughout manuscript.

Line 290. "described" should be "shown".

We have corrected this kind of wrong usages throughout the manuscript.

Line 291-294. "That is, … 3b)." This is very hard to understand.

This paragraph has been rewritten to clarify the changes of wind speed during research period.

Figure 3. It seems like the figure in this data encompasses AWS data for the full country. Correct? This was not introduced in the Data and methods section. How many stations are here? It is impossible to know this since the authors seem to have used some interpolation technique that is also not explained.

Correct, the nationwide AWS data was used. There were 759 AWSs in Korea and their observational parameters were interpolated to given grid based on the objective analysis. We have added detailed descriptions about the characteristic of AWS data (Lines 174-191) and their distributions in Fig. 1 (Line 212).

Line 301. Ambiguous subtitle. Perhaps change to "Upstream environmental conditions?"

Revised as suggestion (Line 373).

Line 306-307. "Three scanning lidars were deployed at …". Three at each site? I know what the authors want to say, but it should be clear from the sentence directly.

This sentence has been rewritten for clarity (Lines 377-379).

Line 307. "Five soundings … coastal area". Something is missing in this sentence.

The missing word has been corrected in this sentence (Line 379).

Line 308. "The sounding … side (GWW)". This should be in methodology section.

This description has been moved to the methodology section (Section 2.2, Lines 197-201)

Line 311. Are BKC and GWW also sounding stations? It is unclear.

Yes, they both are sounding stations, we have modified the description for clarity (Line 200).

Line 316-317. "Furthermore, … symbols." Redundant sentence.
The redundant sentence has been removed.

Line 320. Please remove "Instead … site," as it is redundant information.
The redundant part has been removed.

Lines 331-332. Awkward phrasing.
This sentence has been rewritten for clarity (Lines 397-400).

Figure 4. What is the wind direction in the wind barb plots? Degrees from north, or across the panel? Figure 1a indicates that MOP and JSC are not aligned along this cross section. How is this corrected for? Otherwise, this needs to be acknowledged for somehow: either that data is or is not corrected for the location. Given the WISSDOM dataset doing some interpolation, it seems crucial information at this point of the manuscript. Also, please discard the filled contour for terrain elevation (or make it lighter in color) as it obscures some of the wind barbs at lower elevation.
The wind barb indicates the degree from north, this information has been added in the caption (Lines 404-405). Since the sounding were used here to represent local environmental condition in the scale around 15 km (cf. Lines 204-205). The sounding sites were perpendicularly projected to the cross line (in Fig. 1b) from their original locations, this description has been added in the figure caption as well (Lines 406-407). WISSDOM uses Cartesian coordinate system, thus, the input data have to interpolate to the same coordinate system first, this description has been revised in the introductions of WISSDOM (Section 2.4, Lines 263-265). The filled contour has been removed for clarity (Fig. 4, Line 401).

Line 339. DDG is upstream from the lee slope, but it seems there are more stations even further upstream. Why was this site chosen here?
The DGW is good upstream site than other two sites (MOO and JWC). First, the DGW have no missing data, and it have more tight relations with its downstream site (GWW). These descriptions have been noted in Lines 410-415.

Line 343. Awkard phrasing. It is the air that becomes drier and warmer, not the temperature.
The sentence has been revised (Line 418).

Line 346-347. This is clearly a wrong statement. The authors refer to an elevated

inversion at around 800 hPa in a profile that starts at 900 hPa (Figure 5). Stable boundary layers that develop overnight rarely exceed 300 m agl. Besides, there is clear neutral layer between the surface and the elevated inversion. Thus, this elevated inversion has some other origin, perhaps large-scale subsidence? The authors could address this by simply mentioning that the origin of the elevated inversion at time of writing has not been investigated.

We also agree this point, the large-scale subsident would possibly provide more contributions to the inversion in this event. However, it is not easy to clarify this issue completely from our present analysis and datasets, the numerical study would be good approach. The descriptions about his issue have been improved in manuscript based on the suggestions (Lines 422-426).

Line 348. What is a "good condition" for generating hydraulic jump and downslope windstorm in the lee side? Please be specific.

The "good condition" indicates perpendicularly upstream wind to the mountain range, and upstream inversion. In addition, we modified the word from "good" to "preferred". The specific description about the "preferred condition" has been noted in Lines 426-427.

Line 350. What are environmental winds? Perhaps the authors mean to say "the upstream environment encompassed westerly winds".

Yes, we putted "upstream" prior to "environmental winds" (Line 429).

Line 352. "dramatically". Please remove.

This word has been removed.

Line 358. Headers should be objective titles such as "Lee slope winds" rather than "stronger winds in the lee slope". Please address also for other subtitles.

The subtitles have been revised throughout the manuscript (Line 439, Line 604).

Line 360. Perhaps rephrase to "the prevailing wind direction". Why is this "likely" the wind direction? Wasn't this observed?

This sentence has been revised as suggestion (Line 441).

Line 370. Fluctuations of what?

This sentence has been improved for clarity (Line 450).

Lines 370, 380. What are perturbation temperature and pressure?

We utilized the "station pressure" and "temperature" for further analysis, the descriptions

about the station pressure and temperature have been revised in this paragraph (Line 450 and Lines 460-463).

Figure 6. This is quite a nice figure, but perhaps a lower density in the wind barbs (vertically) would make a clearer picture. Perturbation temperature and pressure. What are these perturbations of? A difference from one-day average at a single station? Or a time-difference across a station-average? Why not just present local ambient and dew point temperature evolution?
Thank you for these good suggestions. The figure has been modified (Line 468) and we also use the station pressure and temperature for further analysis.

Line 395. Which two sites?
It should be "two different locations", the description has been revised for clarity (Line 527).

Line 420-421. As the boundary layer height changes over time, this cannot be a fixed value by definition.
In this budget analysis, the mean of boundary layer height was usually used to represent the $H$. Furthermore, the values of $H$ did not change too much during the research period. Thus, we used these fixed values to represent the boundary layer height here. The description has been modified in Lines 552-555.

Line 433. What is a sub-synoptic scale feature?
It indicates small-scale. The description has been improved for clarity (Lines 567-568).

Line 443. What gusty wind are the authors referring to here?
The gusty wind indicates the wind speed was increased suddenly (like ~3 to 12 m s$^{-1}$ in this event) in short period. This description has been also noted in Lines 579-582 for clarity.

Line 452-453. Perturbation of what?
We used new analysis to evaluate the contributions of large-scale weather system in the PGF based on suggestions below. This paragraph has been rewritten.

Line 458. Surface or sea-level pressure?
This paragraph has been rewritten.

Line 469-470. What would be the rationale between an enhanced PGF and subcloud cooling and/or warming? Wasn't this study performed in fine weather conditions,

meaning there are no clouds involved? Also, just because term B and PGF "trend" overlap, this doesn't necessarily mean that "subcloud warming" is the critical factor explaining the enhance pressure gradient. The "warming" can also come from adiabatic compression as a result of mountain waves involved. In other words, the correlation does not necessarily mean causality here. Is the pressure gradient not just merely a result of the low and high pressure systems going through the area, that with some critical upstream upper-air environment led to some warming down the lee slopes?

We agree this point, and this is particularly an excellent suggestion. Thus, we did new analysis based on this approach. We evaluated the contributions from the large-scale weather system, and the results indicates that relatively lower pressure in the speed-up stage may have produced by adiabatic warming coupled with the LPS. Thus, the large-scale weather system indeed contributed some warming down the lee slope. The detailed descriptions have been revised in last paragraph in Section 4.1.2 (Lines 579-597), and the new figure is shown in Line 599.

Line 504. "Maximum values". What do the authors mean with maximum values?

We replaced the words "maximum values" to become "range". The "range" of Froude number ($F_r = U/NH$) was calculated when we assumed represent terrain height (i.e., $H$ in $F_r$) between 1000 and 2000 m MSL (the averaged altitude of the TMR is ~1200 m). This description has been improved for clarity (Lines 505-509).

Line 506. What sensitivity test are the authors referring to here?

Except for the different represent terrain height, we also calculated the Froude number with different Brunt-Vaisala frequency (including saturated and dry). After this procedure, the range of Froude number in this event were estimated. This description has been improved for clarity (Lines 505-509).

Line 506-507. "increasing the topography height between 1000 and 2000 MSL". What does this mean?

Although averaged height of the TMR is ~1200 m, the Fronde number were estimated by adjusting represent height to check its variability in this local area. In addition, it should be 1000 and 2000 "m" MSL, this typo has been corrected (Line 508).

Line 509-512. It is hard to follow this.

The descriptions have been improved for clarity (Lines 509-512)

Line 513. "surface velocity". Surface winds perhaps?

This word has been corrected (Line 513).

Figure 10. This figure probably provides explanation for the adiabatic compression leading to a warming in the lee slopes. A recommendation is that it would be better to show theta every 2 or even every degree. Figure 10a also raises the question whether rotor behavior was involved or not. By any means, it looks like the development of winds and temperature at GWW could also be influenced by the fact that this location is close to the ocean, which makes the presented analysis a little more tricky. There have been quite a few studies to downslope windstorms in coastal mountainous environments (Corsica, Southern California, Adriatic Sea), maybe have a look at those.

The figure has been modified by following suggestion (Line 521). The rotors seem not clear showing when we increased the contours. Additionally, the ocean temperature was not changed too much during the research period when we checked the sea surface temperature of EAR5. Compared to the downslope windstorms in coastal mountainous environments in the other locations (Corsica, Southern California, Adriatic Sea), we may have assumed that the influences from the ocean would be small in our selected event. The descriptions about this statement have been noted in Lines 516-520.

Figure 10. This is clearly numerical simulations, but the title of the paper says "Observational analysis".

This figure was analyzed by using the LDAPS reanalysis dataset, this reanalysis dataset was assimilated with many of high spatiotemporal wind observations. We have addressed this in the introductions of LDAPS and ERA5 datasets (Lines 220-241). Therefore, the title was modified as well.

Line 523. Please change the title of this section to something that is addressed in the section, rather than "stronger winds".

The title of this subsection has been changed as suggestion (Lines 604-605).

Line 526. "westerly". Please change to "westerly winds". This accounts to all occurrences in the manuscript.

These words have been revised throughout the manuscript.

Line 533. "Can sustain". Please change to "sustained".

The words have been changed as suggestion (Line 614)

Figure 11. Change figure caption to "Same as Figure 6, but for DGW site". See also

comments on figure 6.

The figure caption has been revised (Line 634, Line 661).

Lines 627-629. This is quite an interesting analysis, but this statement seems off. Regarding the minimum and maximum values at roughly 4.5 and 8.5 km, respectively, one sees a similar increase in wind speed of roughly 40%. 10 vs. 15 m/s and 6.5 vs 4.5 m/s. Why, if the wind direction is the same, would this ratio be different in different wind speed conditions?

This is a valid point. This ratio explained that the westerly winds indeed were accelerated in narrow segment along valley, however, there were different amplifications in the maximum wind speed with different strength of westerly. In this event, the maximum wind speed was amplified significantly (~10 m s$^{-1}$ more than averaged) in the narrow segment of valley when the westerly winds were strong. The detailed descriptions have been revised in Lines 695-703.

Figure 14. The y-label says averaged wind speed, but the figure also shows shading. Is that also averaged? Probably it would be better to just change the y-label to wind speed. Does the channel width actually mean valley width?

Thank you for pointing out the problem, the figure has been modified (Line 705).

---

## Author Comment (AC2)

**Responses (highlighted with blue) to Referee #2**
**31 May 2021**

General comment:

The manuscript described two different wind intensification cases in a mountain area from one event, using the reanalysis data, Doppler lidar data, and wind profiler data. The authors performed many analyses using the limited datasets, and the manuscript included many figures. However, the manuscript lacked important descriptions about the novelty of this study, validations of the data used, and the generality of the observed case(s), as commented below.

We appreciate Referee#2's helpful and constructive comments, which help us to improve the manuscript substantially. We have more emphasized the importance and uniqueness of our selected event; the capability and reliability of adopted datasets were also addressed in our revised manuscript. Additionally, the climatological information has been added to understand more about the generality for the event. A set of responses to your comments is provided below and specific locations of revised manuscript were also noted as following responses and revised/added lines in the manuscript.

Major comments:

- If I understand the study correctly, the manuscript described two different wind intensification cases from one event. The first part of the manuscript described characteristics of a wind intensification observed at the lee side of the mountain range, and the latter part described another intensification in a valley. The title of this manuscript included "downslope," therefore, I did not know why the latter part was needed. I could not understand the relationship between the two intensification cases. The introduction also lacked information about the relationship.

Thank you for pointing out the problems. In this extreme strong wind event, there were two different wind patterns (one is more like "gust" in the leeward side of the mountain, another one is "sustained" strong wind over the mountainous area), and they all have huge impacts on human activities (ex: Olympic games and wildfires) in Korea. Furthermore, these two different wind patterns were both presented near Taebeak Mountain Range (TMR) at the same time when a low-pressure system (LPS) was approaching. The results implied that the interactions between large-scale weather systems and complex terrain should play an important role in dominating the wind patterns in this area. We have analyzed the detailed evolution of the winds and their mechanisms have been verified. Although the pressure gradient force (PGF) is main factor to accelerate the winds, different mechanisms were found in these two locations due to different topographic features. A description has been improved (especially in abstract, Lines 45-47) and to emphasize the discrepancies of wind patterns in this mountain range, additionally, the title has been modified for clarity.

- It seems that a large part of the analysis of this study relies on the reanalysis data (LDAPS), which are not pure observational data. Were the observations assimilated enough to resolve the local wind intensifications (time? spatial resolution?)? The authors should provide more details about the dataset and evaluate the dataset in terms of how the data can resolve the local wind intensifications.

    It is a valid point. In this study, we use the reanalysis dataset of the LDAPS but does not used its forecast outputs. Basically, the LDAPS reanalysis dataset was assimilated by various platforms with high resolution of wind observations (like weather radar, AWS, sounding, wind profiler and satellite). The errors between conventional observations and LDAPS have been minimized conscientiously by the KMA and the quality of wind information is able to resolve small-scale weather phenomena over complex terrain in Korea. We modified the title to "**An analysis of an extreme wind event in a clear air condition associated with a low-pressure system during ICE-POP 2018**". The detailed descriptions about the LDAPS dataset have been improved in Section 2.3 (Line 220).

- This manuscript presented a case study of one case only and lacked descriptions/analysis about the generality of the event. How frequent did the wind intensifications happen? Is the mesoscale pressure pattern common? Did the mesoscale pressure pattern always produce the wind intensifications? Are the analyzed phenomena unique in this area?

    This a particularly good suggestion. The uniqueness of our selected event has been addressed based on the historic observational record of KMA. In fact, this event is one of two extreme strong wind events in past decade. The two extreme strong wind events were both affected by the passing LPS (low-pressure system), the result implied that the mesoscale pressure pattern is usually an important factor to dominate the wind intensifications in Korea. The detailed descriptions about the climatological information have been revised in Lines 352-367.

- It was unclear what is the new finding(s) of this study. What are different compared with the previous studies?

This study is first attempt to document the evolution of extreme strong winds in the TMR associated with the moving LPS by observationally cased datasets. Furthermore, the mechanisms of the strong winds over complex terrain have also been verified. Although there were a few numerical studies to investigate the wind patterns and they provided some explanations in the same areas, it still lacks observationally based evidence to convince their findings. This study provided new insights that the PGF play an important role to accelerate the wind speed, however, there were two different mechanisms dominating the PGF associated with the interactions between the LPS and terrain (i.e., channeling effect and adiabatic heating coupled with LPS). We have emphasized these points in the paragraphs of this revision (Lines 102-106, Lines 139-142).

- It was tough to follow the manuscript, because I felt a difficulty to identify a downstream/upstream site through the manuscript. I was confused about which site corresponded to a downstream/upstream in each sentence. I think that major reasons of my confusion are:
- Observation sites had similar names (e.g., DGW, GWW, GWU).

  Since these were officially given names from the KMA, we still prefer to use those names in this study. However, we summarized that information for each station in a table (Line 211) to make it clearer (based on the suggestion from Referee#1).

- Fonts of the site names in each figure are too small.

  The fonts of the site names have been enlarged in all figures throughout the manuscript.

- In the first part of the study, DGW was referred as an upstream site, while it was referred as a downstream site in the later part. I would suggest reorganizing the paragraph or rewording sentences to make the reader easy to identify the downstream/upstream sites.

  The paragraphs and sections have been reorganized and the words (upstream) have been replaced when we mentioned the difference between the YPO and DGW (Section 4.2.2, Line 635).

- It was unclear that how the pressure gradient force was produced by adiabatic warming at the lee side of the mountain. Was there precipitation in the mountain area? It was also unclear that how/why the pressure gradient force was intensified in the DGW site.

  To clarify the causes of PGF in the leeward side of the mountain. We were following a constructive suggestion from Referee#1; relatively lower pressure may have contributed from large-scale weather systems. Therefore, we did a new analysis to evaluate the contributions from the LPS. The results from this new analysis suggested that relatively lower pressure was deduced by the combined effect in speed-up stage. In particularly, the PGF was produced by adiabatic warming coupled with the passing LPS at the leeward side of the mountain, the details about the intensified PGF have been revised in last paragraph in Section 4.1.2 (Lines 579-597) and Figure 10. There are no precipitation along the northeastern coast of South Korea according to the AWS observations. Thus, we can eliminate the effects from the precipitation in this event.

- Language:
- There were so many "can" and "could" used in the manuscript, especially in Section 2. This obfuscates the sentences. I was confused by this, and it was unclear that the things were actually done or not.

  We have checked these problems and corrected them carefully throughout the manuscript.

- Sentences with past forms and those with present forms were mixed inconsistently in the same paragraphs, even in the same sentences.

  The manuscript edited by a professional editorial company to avoid this kind of sentences and to improve readability.

- There were many sentences that used parentheses to state inverse things (e.g., lines 405, 420, and many others). I needed to read the sentences back and forth. This technique should not be used so frequently in a manuscript.

  We have fixed all of these sentences throughout the manuscript.

- "East Sea": Because I did not know "East Sea," I googled it and found that there is "Sea of Japan naming dispute." I recommend using "Sea of Japan," which has been most commonly, historically used in the world, or putting down with "East Sea," like "Sea of Japan (a.k.a. East Sea in Korea)." Alternatively, do not use both names in the manuscript to avoid the unnecessary argue.

We modified to " East Sea in Korea" (Line 328)

Minor comments:

Line 49: GDW was first used here.
Rewords to "a mountainous station" in abstract for clarity (Lines 53-54).

Figure 2: Use consistent formats for x- and y-axis labels.
The format of labels in Figure 2 have been modified (Line 319).

Line 323: What is 184 for?
This redundant word has been removed.

Figure 4: What does the wind burb direction represent? Horizontal wind direction, direction along the cross section, or others?
The tail of wind barbs indicates the wind direction. This description about the wind barbs has been added in the figure caption (Line 404-405).

and 11: Are the labels of terms consistent? I was confused.
They are different. Since the wind speed have significantly different between lower and higher layers, it is difficult to identify the wind speed patterns by the same color. Thus, we preferred to use different color labels in these two figures to emphasize their characteristics.

Line 470 "sub-cloud warming": Did clouds form? Where? Maybe this manuscript needs more descriptions about the weather condition including clouds (and precipitation). Introduction highlighted and stated that wind intensification influences precipitation, while this case did not produce precipitation. This could be a reason that the value of this study was ambiguous.
There were no precipitation in this event, to avoid the confusing and follow the suggestions from the reviewer. We have removed redundant descriptions about the precipitation in introduction and emphasize clear air conditions in this event (Lines 335-338). We have also evaluated the contributions from large-scale weather system. The results indicates that the PGF was produced by adiabatic warming coupled with the passing LPS. The details have been revised in Section 4.1.2 (Lines 579-597).

Lines 366-367: I did not understand how/why the propagation of the upper wind toward the low level was related to the wind intensification.

This sentence was moved to Lines 504-505 to appropriately explain the wind intensifications from the LDAPS dataset.

Figure 11: I did not see any wind intensification near the surface at the DGW site. This is inconsistent with Fig. 13. Why?

There were no significant wind intensifications at the DGW site since it has manifested persistent strong wind when the LPS was approaching. This sustained strong wind produced the PGF associated with channeling effects, and the channeling effect accelerated the wind speed only around the DGW site. This is the reason why DGW always measures stronger wind than the YPO site (this result is consistent with Figure 13).

Line 564: What does "an almost out of phase" mean?

It means that there is a negative relation (opposite phase) between the wind speed and pressure. The wind speed usually increased when the pressure was dropped. The description has been improved in Line 640.

Lines 572-576: I could not understand this sentence. This is too long. Please also check the grammar.

This sentence has been removed due to the new analysis was performed.

Lines 626-630: The sentences did not make sense to me. The second sentence did not follow the third sentence. Maybe need more descriptions.

These sentences have been revised for clarity in Lines 695-704.

Line 651: I could not find the high wind speed area.

The color labels have been changed in Figure 15 (Line 729) to clarify the strong wind speed near the DGW site (narrow segment along the valley).

**An analysis of an extreme wind event in a clear air condition associated with a low-pressure system during ICE-POP 2018**

**Chia-Lun Tsai[1], Kwonil Kim[1], Yu-Chieng Liou[2], Jung-Hoon Kim[3] , YongHee Lee[4], and GyuWon Lee*[1]**

[1]Department of Astronomy and Atmospheric Sciences, Center for Atmospheric REmote sensing (CARE), Kyungpook National University, Daegu, Korea

[2]Department of Atmospheric Sciences, National Central University, Jhongli, Taiwan

[3]School of Earth and Environmental Sciences, Seoul National University, Seoul, Korea

[4]Numerical Modeling Center (NMC), Korea Meteorological Administration, Seoul, Korea

Revised

*Atmospheric Chemistry and Physics*

May 2021

* Corresponding author: Prof. GyuWon Lee, E-mail: gyuwon@knu.ac.kr

**Abstract**

An extreme wind event under clear air conditions on 13–15 February 2018 during the 2018 Winter Olympic and Paralympic games in Pyeongchang, Korea, was examined using various observational datasets and reanalysis data. High spatiotemporal resolution wind information was obtained by Doppler lidars, automatic weather stations (AWSs), a wind profiler, sounding observations, global reanalysis (ERA5) and the local reanalysis datasets from the 3DVAR data assimilation system under the International Collaborative Experiments for Pyeongchang 2018 Olympic and Paralympic winter games (ICE-POP 2018). This study aimed to understand the possible generation mechanisms of localized strong winds across a high mountainous area and on the leeward side of mountains associated with the underlying large-scale pattern of a low-pressure system (LPS). The evolution of surface winds shows quite different patterns, exhibiting 1) intensification of strong winds in the leeward side and 2) persistent strong winds in upstream mountainous areas with the approaching LPS. The two different mechanisms of strong winds were investigated. The surface wind speed was intensified dramatically from ~3 to ~12 m s$^{-1}$ (gusts were stronger than 20 m s$^{-1}$ above the ground) at a surface station in the leeward side of the mountain range. A budget analysis of the horizontal momentum equation suggested that the pressure gradient force (PGF) contributed from adiabatic warming and the passage of LPS was the main factor in the dramatic acceleration of the surface wind in the downslope, leeward side of the mountains. However, a mountainous station appeared to have persistent strong winds (~10 m s$^{-1}$). Detailed analysis of the retrieved 3D winds revealed that the PGF also dominated at the mountainous station, which caused persistent strong winds related to the channeling effect along the narrow segment of the valley in the mountainous area. The observational evidence showed that under the same synoptic condition of a LPS, different mechanisms are important for strong winds in this local areas in determining the strength and persistence of orographic-induced strong winds under clear air conditions.

**1. Introduction**

Wind is an important atmospheric phenomenon, and topography can significantly affect the behavior of winds to accelerate/decelerate the wind speed or to change the wind direction (Mitchell, 1956; Brinkmann, 1974; Houze, 2012; Yu and Tsai, 2017; Tsai et al., 2018). Such orographically strong wind and mountain waves can easily induce very large impacts on aviation operations (Clark et al., 2000; Kim and Chun, 2010, 2011; Kim et al., 2019; Park et al., 2016, 2019), outdoor sport activities, and forest wildfires in a relatively dry environment under clear air conditions (Smith et al., 2018). Downslope windstorms can produce strong winds on the leeward side and play an essential role in creating and maintaining wildfires near northern California with easterly winds across the Sierra Nevada and southern Cascade Mountains (Mass and Ovens, 2019). Lee et al. (2020) also suggested that downslope windstorms favor wildfires along the northeastern coast of Korea with westerly winds across the Taebeak Mountain Range (TMR). In addition, wind speeds are also usually accelerated locally near narrow valleys or channels between mountains, such as the "gap winds" occurring along the strait of Juan de Fuca in Washington (Reed, 1931; Colle and Mass, 2000), Columbia River Gorge in Oregon (Sharp, 2002), and Jangjeon area in South Korea (Lee et al., 2020).

The environmental conditions of large-scale weather systems are key factors in determining the locations where strong winds are generated. Downslope windstorms usually occur on the leeward side of a mountain range, and the upstream prevailing wind direction is mostly perpendicular to the orientation of the mountain range. An elevated inversion layer and the height of the mean-state critical level are also important references to evaluate the occurrence of downslope windstorms. The occurrence of downslope windstorms are usually accompanied with hydraulic jumps, partial reflection, and critical-level reflection, according to various numerical and theoretical studies in the past few decades (Long, 1953; Houghton and Kasahara, 1968; Klemp and Lilly, 1975; Smith, 1985; Durran, 1990; Afanasyev and Peltier, 1998; Epifanio and

Qian, 2008; Rögnvaldsson et al., 2011; Cao and Fovell, 2016). The combination of hydraulic jumps and wave breaking can also enhance downslope windstorms and increase the wind speed (Shestakova et al., 2018; Tollinger et al., 2019). The pressure gradient force (PGF) is one of the possible factors that accelerates the wind speed near the exit of the gap between the mountains when prevailing winds blow into a narrow valley with appropriate directions (Reed, 1931;

Finnigan et al., 1994; Colle and Mass, 2000). Although the characteristics of these two kinds of orographically strong winds (downslope windstorms and gap winds) are fundamentally different, they may occur on adjacent mountains at the same time (Hughes and Hall, 2010; Lee et al., 2020).

A few previous numerical studies have provided insightful explanations about the development of the strong winds associated with the downslope windstorms along the northeastern coast of South Korea (on the leeward side of the TMR). Most of the strong downslope wind events were mainly explained by the three mechanisms in this region: hydraulic jump, partial reflection, and critical-level reflection (Lee, 2003; Kim and Cheong, 2006; Jang and

Chun, 2008; Lee and In, 2009). Strong winds can occur during any season with the appropriate environmental conditions, such as westerly winds and upstream inversion. Lee et al. (2020)

confirmed these conclusions with numerical modeling studies. Furthermore, they also found that the PGF is one of the possible factors to produce the gap wind, and the variability of the PGF is highly related to the local topographic features. However, sufficient observational studies to examine the detailed mechanisms of orographically-induced strong winds and their relations with large-scale weather systems in Korea are still lacking because relatively dense wind observations from ground-based remote sensing techniques cannot be easily collected under clear air conditions.

Pyeongchang hosted the Winter Olympic and Paralympic Games in 2018 (most venues were located in coastal and higher elevation areas of the TMR). More detailed weather conditions and accurate prediction for several key parameters, such as precipitation, visibility, wind directions, and wind speed, are important to ensure the safety of all athletes and attendees. The Numerical

Modeling Center (NWC) of the Korea Meteorological Administration (KMA) organized an intensive field experiment named the International Collaborative Experiments for Pyeongchang 2018 Olympic and Paralympic winter games, ICE-POP 2018 (http://155.230.157.230:8080/Icepop_2018/index.jsp). A very dense observational network was built to provide a high-quality observational dataset at high temporal and spatial resolutions under either precipitation or clear air conditions. Many kinds of instruments were involved in ICE-POP 2018, which allows the observationally based investigation of the nature of the strong wind event in the nearby mountainous area.

Scanning Doppler lidar can be one approach to obtain more complete wind information in such conditions with even finer resolutions. A few studies have used Doppler lidar to document orographic flow, downslope windstorms and rotors (Neiman et al., 1988; Hill et al., 2010; Mole et al., 2017; Bell et al., 2020). Kühnlein et al. (2013) found that transient internal hydraulic jumps are characterized by turbulence. Menke et al. (2019) identified the recirculation zone over an area with complex terrain using six scanning Doppler lidars. The interactions between the winds and terrain dominantly affected the occurrence of flow recirculation. However, only radial winds were used, resulting in incomplete wind observations that can provide only limited information for realistic airflow structures. Complete 3D wind fields could be retrieved from 4D-Variational Assimilation (4DVAR) using Doppler lidar. The accuracy of wind speed, direction and water vapor flux are improved when assimilating lidar data (Kawabata, 2014). Thus, lidar observations can indeed provide high-quality 3D wind information under clear air conditions.

The objective of this study is to use high spatiotemporal resolution datasets to investigate the fine-scale structural evolution of strong winds over the complex terrain in the northeastern part of South Korea (i.e., in the Pyeongchang area) during 13–15 February 2018. Multiple Doppler lidars, automatic weather stations (AWSs), a wind profiler, sounding observations, global reanalysis (ERA5) and the local reanalysis datasets from the 3DVAR data assimilation system (LDAPS: Local Data Assimilation and Prediction System) were adopted to analyze the detailed wind patterns over the TMR and northeastern coastal regions. The 3D winds were also derived through the WInd Synthesis System using DOppler Measurements (WISSDOM, Liou and Chang,

2009; Tsai et al., 2018) synthesis. Since only a few extreme wind events were identified here based on the KMA historic record in the past decade (see details in Section 3.1), the impact of large-scale weather systems on triggering strong winds over complex terrain is still unclear, especially under clear air conditions. Therefore, this study is the first observationally based attempt to recognize the mechanisms of the strong winds over the TMR while a low-pressure system (LPS) passes through the northern side of the Korean Peninsula. A unique extreme wind event was selected for further analysis not only because the Olympic games were interrupted due to the strong wind invading the mountainous area and leeward side of the mountain range but also because dense observations are available during ICE-POP 2018. Furthermore, three scanning

Doppler lidars were established in this area, which provided more sufficient wind information under clear air conditions.

**2. Data and methodology**

**2.1 Scanning Doppler lidar**

Two different models of scanning Doppler lidars were adopted in this study: (1) "WINDEX-

2000" produced by the manufacturer Laser Systems and (2) the "Stream Line" produced by the manufacturer HALO Photonics. The scanning Doppler lidar measures the radial Doppler velocity by detecting atmospheric aerosols and dust via a laser (class 1 M) at an exceedingly high spatial resolution. The radial winds were sufficiently observed by an adjustable scanning strategy in three modes: plan position indicator (PPI), range height indicator (RHI), and zenith pointing (ZP).

Furthermore, these lidar observations were used to construct the complete wind information under clear air conditions via WISSDOM.

The WINDEX-2000 lidar operated a full volume scan every ~27 min with seven PPIs (elevation angles of 5º, 7º, 10º, 15º, 30º, 45º, and 80º) and one hemispheric RHI (azimuth angle of 0º, that is, starting from the north). There are 344 gates along a lidar radial direction with 360

azimuth angles between 0º and 360º. The gate spacing is 40 m, and the maximum observed radius distance is ~13 km. The Stream Line lidar operated a full volume scan every ~13 min with five

PPIs (7º, 15º, 30º, 45º, and 80º before 10:00 UTC on 14 Feb. 2018 and 4º, 8º, 14º, 25º, and 80º

after 10:00 UTC) and two hemispheric RHIs (azimuth angles of 51º and 330º). There are 1660

gates along the 360 lidar beams with azimuth angles between 0º and 360º.

Quality control (QC) of the radial winds (in PPI and RHI modes) was performed by applying the signal noise ratio (SNR) threshold in advance. To obtain correct and useful measurements, QC is necessary for each lidar observation, where the nonmeteorological echoes are removed when the SNR threshold is smaller than 0.04. This threshold was obtained by a series of tests, and it can appropriately remove most of the noise and retain sufficient meteorological signals at the same time.

**2.2 Automatic weather stations (AWSs), soundings, and the wind profiler**

Fig. 1 shows the main study domain (larger box in Fig. 1a), WISSDOM domain (box in Fig.

1b) and domain of mountain clusters during ICE-POP 2018. The locations of all AWSs are also marked in Fig. 1a. There were 727 regular operational stations, and the mean distance between

AWS stations was ~10 km. Two distinct dense areas of the AWS observations were found: one was located near Seoul city (~37.5°N, 126.7ºE), and the other was located inside the smaller box over the TMR. This is because additional AWS sites (32 stations) were deployed in the mountainous area during ICE-POP 2018. This dense AWS network (black dots in Fig. 1) is utilized to document the detailed evolution of surface parameters and as one of the constraints in

WISSDOM (details are given in the following subsection). The AWSs mainly provide the surface wind speed, wind direction, pressure, and temperature at high temporal resolution (1-min interval). Original AWS observations reveal semirandom distribution and must be interpolated on given grids in a Cartesian coordinate system after applying objective analysis (Cressman,

1959) with a suitable influence of radius (10 km in this study). These gridded AWS data will be of great benefit to WISSDOM and further analysis of wind speed changes in Korea. Note that three AWS stations were selected to represent the fluctuations of pressure, temperature and winds in the mountainous areas (YPO and DGW sites) and the leeward side of the mountain range (GWW site).

There is only one regular sounding station (GWW) inside the main study domain operated by the KMA twice a day (00Z and 12Z); such a coarse dataset is quite limited for representing the local changes in environmental conditions near the TMR. Therefore, four additional soundings (DGW, BKC, JSC, MOO) were launched every 3 hours (from 00Z) during the research period (except for the JSC site, see Table 1 for details), and the sounding sites were located inside the study domain near the northeastern part of South Korea (black squares in Figs. 1b and 1c). The

MOO and JSC sounding stations were located in the southwestern TMR with a gentle slope, and the DGW station was the closest site to most outdoor venues of the Olympic games near the crest of the TMR. The other two sounding stations, BKC and GWW, were located on the northeastern slope of the TMR and in the coastal area, respectively (Fig. 1b). The sounding observations provide detailed horizontal winds, temperature profiles (~1 m vertical resolution), and stability information across the mountainous and coastal areas. Such dense sounding observations are adequate to represent the local environmental conditions on a relatively small scale (~15 km) in the study domain when the LPS passed through.

A wind profiler was deployed at the GWW site to measure the winds in the case of a lack of sounding observations. In addition, the high temporal resolution of wind profiler measurements (10-min interval) could potentially be a reference for the surface and retrieved winds. The names of adopted sites, their equipped instruments and temporal resolutions are summarized in Table.

Additionally, intensive observations during ICE-POP 2018 are marked by asterisks.

Table 1 General information of the observational sites

| Observation site | Operating instrument(s) | Temporal resolution | Location | Elevation (m, MSL) |
|---|---|---|---|---|
| DGW | Lidar (Stream Line)*
 Sounding*
 AWS (#100) | 13 mins
hours
min | 37.677°N,128.718°E (mountainous site) | 773 |
| MHS | Lidar (WINDEX-2000)* | 27 mins | 37.665°N,128.699°E (mountainous site) | 789 |
| GWW | Wind Profiler
 Sounding
 AWS(#104) | 10 mins
hours
min | 37.804°N,128.854°E (leeward side) | 79 |
| GWU | Lidar (Stream Line)* | 13 mins | 37.770°N,128.866°E (leeward side) | 36 |
| BKC | Sounding* | 3 hours | 37.738°N,128.805°E (leeward side slope) | 175 |
| JSC | Sounding* | 3-6 hours | 37.475°N,128.610°E (mountainous site) | 424 |
| MOO | Sounding* | 3 hours | 37.562°N,128.371°E (mountainous site) | 532 |
| YPO | AWS(#318) | 1 min | 37.643°N,128.670°E (mountainous site) | 772 |

*operated only during ICE-POP 2018

[Figure]

[Figure]

**Figure 1.** (a) Observation sites used in this study and the topographic features (color shading) from the digital elevation model (DEM) in Korea; the arrows mark the location of the TMR. (b) The study domain corresponding to the large box in Fig. 1a was chosen in this study. (c) The WISSDOM synthesis domain adopted in this study corresponds to the small box in Fig. 1a. The locations of the scanning Doppler lidar sites are denoted by asterisks. The locations of the sounding sites are denoted by squares. Note that the sounding and lidar observations are both operated at the DGW site and that a wind profiler is located at the GWW site. The locations of the AWS sites and LDAPS grids are denoted by dots and plus symbols, respectively.

**2.3 Reanalysis data: LDAPS and ERA5**

Generally, LDAPS is a 3DVAR numerical weather prediction (NWP) product generated by the KMA with a spatial resolution of ~1.5 km and temporal resolution of 3 hours with 70 vertical levels. The local reanalysis dataset of LDAPS was used here for further analysis, and the LDAPS

forecast outputs were not included in this study. Various observations were assimilated in this reanalysis dataset to be the initial conditions of LDAPS, and those observations included the

AWS, sounding, wind profiler, radar, buoy, satellite (polar orbit and geostationary equatorial orbit), and aircraft (research and commercial) data. These observational platforms provided highquality and high spatiotemporal resolution wind observations (especially from the AWSs, radar and satellites) for the LDAPS reanalysis dataset, and the error between observations and this reanalysis dataset was sufficiently minimized after careful corrections from the KMA. Such initial conditions have also significantly improved LDAPS forecasting ability of small-scale weather phenomena over complex terrain in Korea (Kim et al., 2019; Choi et al., 2020; Kim et al., 2020). The wind fields from LDAPS are used as one of the constraints in WISSDOM to minimize the errors of retrieved 3D winds and to compare the discrepancies of winds with previous numerical studies (Section 4.1.1). This dataset is freely available from the KMA website (https://data.kma.go.kr).

The ERA5 reanalysis dataset is an atmospheric reanalysis of the global climate and was generated by the European Centre for Medium-Range Weather Forecasts (ECMWF). ERA5 is the fifth generation ECMWF reanalysis with a combination of model and observations. ERA5

provides winds in regular latitude-longitude grid data at $0.25° \times 0.25°$ and 37 pressure levels between 1000 and 1 hPa every hour from 1979 to the present (DOI: 10.24381/cds.adbb2d47).

**2.4 WInd Synthesis System using DOppler Measurements (WISSDOM)**

WISSDOM was originally developed by Liou and Chang (2009) and has been applied in the

Pyeongchang area (Tsai et al., 2018). This study adopted a newly improved version, which includes more observations as constraints compared with a previous version. In the new version of WISSDOM, the following cost function [eq. (1)] is minimized by using a mathematical variational-based method at the retrieval time:

$$J = \sum_{M=1}^{8} J_M. \tag{1}$$

This cost function comprises eight constraints, and the 3D wind fields are obtained by variationally adjusting solutions to simultaneously satisfy those constraints at the same time. The first constraint is the geometric relation between the radial velocity $(V_r)$ observations from multiple lidars and Cartesian winds $V_t = (u_t, v_t, w_t)$, which are control variables, defined as

$$J_1 = \sum_{t=1}^{2} \sum_{x,y,z} \sum_{i=1}^{N} \alpha_{1,i} \left(T_{1,i,t}\right)^2, \tag{2a}$$

$$T_{1,i,t} = (V_r)_{i,t} - \frac{(x - P_x^i)}{r_i} u_t - \frac{(y - P_y^i)}{r_i} v_t - \frac{(z - P_z^i)}{r_i} (w_t - W_{T,t}), \text{ and} \tag{2b}$$

$$r_i = \sqrt{\left(x - P_x^i\right)^2 + \left(y - P_y^i\right)^2 + (z - P_z^i)^2}. \tag{2c}$$

Any numbers of lidar [subscripts $i$ in eq. (2a)] can be applied to this constraint at two time levels (subscripts $t$). $\alpha_1$ in eq. (2a) is the weighting coefficient corresponding to $J_1$ (which is the same in the following equations for $J_2$ - $J_8$). The subscripts $i$ and $t$ in $(V_r)_{i,t}$ represent the radial velocity observed by the $i$-th lidar, $(u_t, v_t, w_t)$ indicate the 3D wind at location $(x, y, z)$, and the terminal velocity $(W_{T,t})$ of particles is estimated by radar reflectivity at two time levels.

$(P_x^i, P_y^i, P_z^i)$ are the coordinates of the $i$-th lidar, and the distance between each grid point and the $i$-th lidar is denoted by $r_i$. Note that $W_{T,t}$ is zero when there is no radar reflectivity, or the terminal velocity is possibly negligible under clear air conditions. Furthermore, all observational inputs (i.e., lidar radial winds, AWS, sounding and LDAPS horizontal winds) must be bilinearly interpolated to given grids in a Cartesian coordinate system before running WISSDOM.

The next constraint is the difference between $\mathbf{V}_t$ and the background winds $(\mathbf{V}_{B,t})$ defined in eq. (3)

$$J_2 = \sum_{t=1}^{2} \sum_{x,y,z} \alpha_2 (\mathbf{V}_t - \mathbf{V}_{B,t})^2. \tag{3}$$

The sounding observations are used as the background winds in eq. (3). The constraint of the anelastic continuity equation is

$$J_3 = \sum_{t=1}^{2} \sum_{x,y,z} \alpha_3 \left[ \frac{\partial(\rho_0 u_t)}{\partial x} + \frac{\partial(\rho_0 v_t)}{\partial y} + \frac{\partial(\rho_0 w_t)}{\partial z} \right]^2, \tag{4}$$

where $\rho_0$ is the air density. The fourth constraint was deduced from the vertical vorticity equation given by

$$J_4 = \sum_{x,y,z} \alpha_4 \left\{ \frac{\partial \xi}{\partial t} + \overline{\left[ u \frac{\partial \xi}{\partial x} + v \frac{\partial \xi}{\partial y} + w \frac{\partial \xi}{\partial z} + (\xi + f)\left( \frac{\partial u}{\partial x} + \frac{\partial v}{\partial y} \right) + \left( \frac{\partial w}{\partial x} \frac{\partial v}{\partial y} - \frac{\partial w}{\partial y} \frac{\partial u}{\partial z} \right) \right]} \right\}^2, \tag{5}$$

where $f$ indicates the Coriolis parameter and the meaning of the overbar in eq. (5) is the temporal average of the two time levels. The constraint of the Laplacian smoothing filter is

$$J_5 = \sum_{t=1}^{2} \sum_{x,y,z} \alpha_5 [\nabla^2 (u_t + v_t + w_t)]^2. \tag{6}$$

The horizontal winds observed by the soundings, AWSs and LDAPS, can be interpolated to each given grid in the WISSDOM synthesis domain. The sixth constraint is the difference between the $V_t$ and the sounding observations ($V_{S,t}$), as defined in eq. (7):

$$J_6 = \sum_{t=1}^{2} \sum_{x,y,z} \alpha_6 (V_t - V_{S,t})^2. \tag{7}$$

The seventh constraint represents the discrepancy between the retrieved winds and AWS ($V_{A,t}$), as expressed in eq. (8):

$$J_7 = \sum_{t=1}^{2} \sum_{x,y,z} \alpha_7 (V_t - V_{A,t})^2. \tag{8}$$

Finally, the eighth constraint measures the squared errors between the horizontal winds and the

LDAPS ($V_{L,t}$), as defined in eq. (9):

$$J_8 = \sum_{t=1}^{2} \sum_{x,y,z} \alpha_8 (V_t - V_{L,t})^2. \tag{9}$$

The main purpose of this constraint is to minimize the squared errors between the horizontal winds of LDAPS and synthesis winds of WISSDOM, which improves the accuracy of retrieved winds. A relatively weak weighting of the LDAPS reanalysis dataset was applied in the

WISSDOM synthesis because more emphasis on the contributions from the other observations is preferred in this study.

The original version of WISSDOM is used only in the case of rain or snow with the first five constraints; it has already been comprehensively applied to synthesize high-quality 3D winds in some previous studies. The retrieved 3D winds consistently revealed reasonable patterns compared with conventional observations or observing system simulation experiment (OSSE)- type tests (Liou and Chang, 2009; Liou et al., 2012, 2013, 2014, 2016; Lee et al., 2017; Chen,

2019). Chen (2019) concluded that the retrieved 3D winds show good relations with observations in several typhoon cases (the mean correlation coefficient was from 0.56 to 0.86, and the root mean square deviation was between 1.13 and 1.74 m s$^{-1}$). The primary advantages and additional details of WISSDOM can be found in Tsai et al. (2018). The main improvement of the new version of WISSDOM is that all available wind observations are considered as one of the constraints to minimize the cost function. In addition, this new version extends its applicability by including multiple-lidar observations and thus, realistic wind fields can be retrieved under clear air conditions.

**3. Overview of the extreme wind event**

**3.1 Synoptic conditions**

The hourly ERA5 dataset was used here to document the synoptic conditions. At the beginning of the research period at 12:00 UTC on 13 February 2018, a high-pressure system (HPS) was located in the southernmost Korean Peninsula (as shown in Fig. 2a). Surface southwesterly winds were dominant from the Yellow Sea to the western coast of South Korea associated with the anticyclonic circulation of the HPS. The southwesterly winds were also related to the cyclonic circulation of a LPS centered at 39$^{\circ}$N, 117$^{\circ}$E near Beijing, China.

Compared to the winds over the western coast, relatively weak winds existed over the land and eastern coast of Korea. The westerly wind came from China accompanying warm air in a higher layer (850 hPa, Fig. 2b). This veering wind also indicated that the prevailing southwesterly wind was dominated by warm advection. Thus, a temperature gradient existed between the land and the western and eastern coasts (exceeding an ~4K difference).

[Figure]

**Figure 2.** (a) Horizontal winds (vectors) and pressure (hPa, color shading) at the surface level, and (b) horizontal winds (vectors) and temperature (K, color shading) at the 850 hPa level obtained from the ERA5 reanalysis dataset at 12:00 UTC on 13 Feb. (c) and (d) Same as (a) and (b) but at 03:00 UTC on 14 Feb. (e) and (f) Same as (a) and (b) but at 00:00 UTC on 15 Feb. 2018. The location of the low pressure system is marked by "L", and the location of the high pressure system is marked by "H".

Consequently, the LPS and HPS were both moving eastward. The surface wind became stronger and turned to westerly winds over the Korean Peninsula associated with the confluences between these two systems (Fig. 2c). The horizontal pressure gradient intensified along the northeastern coast of Korea as the LPS moved to the East Sea in Korea. A relatively low temperature was detected over the mountainous area (i.e., near the northeastern coast of South

Korea), even when the warm advection was approaching Korea (Fig. 2d). Another HPS was moving out from the northeastern coast of China (~40°N, 120ºE) at approximately 00:00 UTC

on 15 February 2018 and the environmental winds surrounding Korea switched to relatively weak northwesterly or northeasterly winds over land at the surface (Fig. 2e). Relatively weak pressure gradients and small temperature differences between the western and eastern coasts are shown in

Figs. 2e and 2f. Since there was no precipitation along the northeastern coast of South Korea according to the AWS observations during the research period (not shown), the lidar observations certainly had the most complete coverages in the study domain without significant attenuations from precipitation particles in this event.

The evolution of surface wind speed observed from all AWS stations over the Korean

Peninsula is shown as a sequence of figures in Fig. 3. At the beginning of the research period, the observed wind speeds were weak in most areas of Korea except for an area near the TMR (Figs.

3a, 3b, and 3c). The surface wind speed was intensified in most areas of Korea when the LPS was approaching at approximately 03:00 UTC on 14 February 2018 (Figs. 3d, 3e, and 3f) and weakened when the LPS moved away from Korea (Figs. 3g, 3h, and 3i). Two distinct wind speed patterns were clearly identified as sustained strong wind speed existed along the TMR and was even stronger in some local mountainous areas from 12:00 UTC on 13 to 14 February 2018, and the strongest surface winds (exceeding ~10 m s$^{-1}$) occurred along the northeastern coast of Korea during a shorter period (approximately 03:00 to 06:00 UTC on 14 February 2018). Since strong winds occurred during both day and night, the changes in surface winds were mainly affected by the interactions between the movement of synoptic weather systems and complex terrain (cf. Fig.

2) and manifested relatively weak relations with diurnal effects in this event.

According to the KMA historic record in the past decade during winter seasons (December to March, 2010-2019), the total number of days with daily maximum wind speeds larger than 10

m s$^{-1}$ is 299 days at DGW and only 19 days at the YPO site. This result indicates that persistent strong winds usually occurred at certain locations over the TMR, such as the DGW site. Although the DGW and YPO sites are both located in mountainous areas with similar elevations and environments (~10 km distance between these two sites), stronger winds are always measured at the DGW site compared to those measured at the YPO site. On the leeward side of the TMR, there were six strong wind events during winter in the past decade based on the KMA AWS

measurements (daily maximum wind speed larger than 10 m s$^{-1}$) at the GWW site. Furthermore, there were only two extreme wind events (wind gusts over 20 m s$^{-1}$) in the past decade at the same site. In these two extreme wind events, their synoptic conditions were both mainly dominated by LPSs. These historical records imply that the frequency of extreme wind events in this local area is highly related to LPSs. One of these two extreme events was chosen, and this unique extreme wind event allowed us to investigate the mechanisms of persistent strong and gusty winds across the mountainous area and the leeward side of the mountain range associated with the influences of LPS movement.

[Figure]

**Figure 3.** (a) Surface wind speed (m s$^{-1}$, color shading) calculated from all automatic weather station observations when the low-pressure system passed through the Korean Peninsula at (a) 12:00, (b) 18:00 and (c) 21:00 UTC on 13 Feb. 2018; (d) 00:00, (e) 03:00, (f) 06:00, (g) 12:00 and (h) 18:00 UTC on 14 Feb. 2018; and (i) 00:00 UTC on 15 Feb. 2018.

**3.2. Upstream environmental conditions in the local area near northeastern Korea**

Because the evolution of surface wind speed revealed quite different patterns in the mountainous area and on the leeward side of the TMR, two domains were selected in this study, which are shown as boxes in Figs. 1a and Fig. 3. All available observations during the intensive observation period are also marked in Fig. 1b and 1c. One type of scanning Doppler lidar was deployed at DGW and GWU (Stream Line), and the other type was deployed at MHS (WINDEX-2000), indicated by asterisks in Fig. 1b. Five sounding stations are aligned from the mountainous area to the coastal area (i.e., perpendicular to the orientation of the TMR). In addition, a wind profiler is located on the leeward side (GWW). The WISSDOM synthesis domain was set over the mountainous area with a horizontal spatial coverage of $12 \times 12$ km$^2$, as shown in Fig. 1c. The horizontal and vertical grid sizes were both set to 50 m, and the vertical extent was from 0 km to 3 km height mean sea level (MSL). Additional AWS stations were deployed around the venues (black dots in Fig. 1c) during ICE-POP 2018.

Fig. 4 shows the variations in the environmental winds observed by the soundings and/or wind profiler along the crossline (black line in Fig. 1b) from the mountainous area to the leeward side of the mountain range. The wind profiler observations are used to provide wind information near the coastal area when the LPS was passing Korea. At the beginning of the research period, prevailing westerly winds were dominant at all sounding sites (Fig. 4a). However, stronger winds were measured at heights below ~1.5 km at only the DGW site near the crest of the TMR (~25 m s$^{-1}$), and weaker winds (<15 m s$^{-1}$) were observed in the lower layers at other sites (MOO, BKC, and GWW) on both the windward slope and leeward side. The wind direction was still westerly at 03:00 UTC on 14 February 2018 (Fig. 4b). Strong winds were detected at the DGW site and downslope with wind speeds larger than 20 m s$^{-1}$ above the BKC and GWW sites. Although the wind speed became stronger above 1.5 km MSL over the DGW site, it did not exhibit a significant change near the surface. These results demonstrate that persistent strong winds existed over the mountainous area (i.e., near the DGW site) while the LPS was approaching. The wind became weak over the mountainous area and leeward side of the mountain range when the LPS moved away from the Korean Peninsula (Fig. 4c).

[Figure]

**Figure 4.** Horizontal winds observed by sounding and wind profiler along the cross line corresponding to Fig. 1b at (a) 12:00 UTC on 13 Feb., (b) 03:00 UTC on 14 Feb., and (c) 00:00 UTC on 15 Feb. 2018. A full wind barb corresponds to 5 m s$^{-1}$; a half barb corresponds to 2.5 m s$^{-1}$. The tail of wind barbs indicates the wind direction (degrees clockwise from the north). The color indicates the wind speed corresponding to the color bars. The thick black line in the lower portion indicates the averaged topography along the line in Fig. 1b. The sounding sites were perpendicularly projected to the cross line from their original locations and are marked in this figure.

Detailed environmental conditions upstream of the leeward site in the TMR (i.e., the mountainous area for westerly winds) were investigated by sounding observations at the DGW site (Fig. 5). Note that the DGW sounding site was selected here to explain upstream environmental conditions because the wind observations seem to have relatively weak relations between farther upstream (MOO) and the GWW sounding site. Unlike the DGW site, the MOO site exhibited unchanging wind patterns when the LPS was passing (cf. Figs. 4a and 4c). Additionally, the JSC sounding site lacks wind observations at a critical time step (12:00 UTC on 13 February 2018). An inversion layer existed at a height of approximately 800 hPa mainly due to the warm advection accompanied by the southwesterly winds at 850 hPa ahead of the LPS

(Figs. 2a and 2b at 12:00 UTC on 13 Feb. 2018) until it passed through the Korean Peninsula (at

03:00 UTC on 14 Feb. 2018). The air temperature increased near the surface and became drier above the inversion layer between the two time steps. The wind direction was westerly at all levels while the LPS passed through. The wind speed became stronger above the inversion layer, but it exhibited no clear changes below ~800 hPa. It is worth mentioning that the inversion layer probably developed due to several factors: (1) large-scale warm advection, (2) a stable boundary layer and (3) large-scale subsidence. However, stable boundary layer is not easily developed at higher levels overnight, and environmental conditions are more like neutral in this event. Thus, determining the separate contributions of these three factors will require a modeling study for this event in the future. The sounding observations showed preferred conditions (i.e., upstream wind direction perpendicular to the mountain range, and upstream inversion) conductive to generating hydraulic jumps and downslope windstorms on the leeward side (Lee et al. 2020).

In summary, the upstream environmental winds associated with the LPS were mostly westerly in this event. However, the wind speeds revealed different characteristics across the TMR, as strong winds (~10 m s$^{-1}$) persisted near the surface in the mountainous area and the wind speed increased on the leeward side of the mountain range.

[Figure]

**Figure 5.** Profiles of temperature (solid lines), dew point (dashed lines), and horizontal winds observed by sounding at the DGW site at 12:00 UTC on 13 Feb. (black lines) and 03:00 UTC on 14 Feb. (red lines) 2018. A full wind barb corresponds to 5 m s$^{-1}$ and a half barb corresponds to 2.5 m s$^{-1}$. The tail of wind barbs indicates the wind direction (degrees clockwise from the north).

**4 Results**

**4.1 Leeward downslope winds**

**4.1.1 A dramatic acceleration of downslope winds**

Although the prevailing wind direction was westerly, the wind speed had a dramatic increase on the leeward side of the TMR. The detailed wind speed and surface fluctuations were documented by a lidar quasi-vertical profile (QVP, Ryzhkov et al., 2016) at the GWU site (upper panel) and the AWS observations at the GWW site (lower panel), as shown in Fig. 6. The wind speed was relatively weak at approximately 6–9 m s$^{-1}$ in the lowest layer at the beginning of the research period. Strong winds were then measured by the lidar QVP reaching ~36 m s$^{-1}$ up to

~1.5 km MSL after 00:00 UTC on 14 February 2018 (Fig. 6a). Finally, the wind speed became weak after 09:00 UTC on 14 February. Winds observed from the sounding and wind profiler were consistent with these QVP winds (cf. Fig. 4).

Fluctuations in surface observations of wind speed, direction, station pressure and temperature at the GWW site are shown in Fig. 6b. The changes in wind speed were similar to the lowest layer of lidar observations (cf. Fig. 6a). Relatively weak winds were measured at the early stage of the period, and the surface wind speed intensified dramatically, exceeding ~12 m s$^{-1}$ between 00:00 and 06:00 UTC on 14 February (named the speed-up stage and highlighted by the shaded area in Fig. 6). The surface wind direction also showed similar patterns to the lidar observations, as it had minor changes from more southerly to westerly. Although these two stations were at different locations along the northeastern coast of Korea, they revealed consistent changes in wind fields. The results also implied that the wind fields along the coast and on the leeward side of the TMR have almost the same characteristics, which could be verified by the analysis of the surface wind speed (cf. Fig. 3f). A relatively low temperature was measured within the first 12 hours at the beginning of the period, and the temperature increased after 00:00 UTC

on 14 February from ~3ºC to 9ºC. The fluctuation in station pressure showed an opposite phase with the temperature variations and the magnitude changed from approximately 1008 to 998 hPa.

The wind speed increased just after the temperature rose and station pressure dropped. That is, a significant lag between changes in temperature, station pressure and wind speed is evident. Their specific relationships and mechanisms are clarified through a more detailed analysis in next

Section.

[Figure]

**Figure 6.** (a) Time series of quasi-vertical profile (QVP) from lidar observations at the GWU site during 12:00 UTC on 13 Feb. to 12:00 UTC on 14 Feb. 2018. A full wind barb corresponds to 5 m s$^{-1}$; a half barb corresponds to 2.5 m s$^{-1}$ and the color indicates the wind speed (m s$^{-1}$) corresponding to the color scale. The tail of wind barbs indicates the wind direction (degrees clockwise from the north). (b) Time series of horizontal winds (wind barbs), wind speed (m s$^{-1}$, black line), station pressure (hPa, P, blue line) and temperature (ºC, T, red line) observed from the AWS at the GWW site. The time period with accelerating wind speed is also highlighted by light yellow shading (i.e., speed-up stage).

To understand more about the discrepancies in downslope windstorm characteristics from previous numerical studies (Lee, 2003; Kim and Cheong, 2006; Jang and Chun, 2008; Lee and

In, 2009). Fig. 7 shows the detailed wind fields and the mountain wave feature that were detected in the local reanalysis dataset of LDAPS. Alternating downdrafts and updrafts were present near the crest (near the DGW site) and leeward side of the TMR at 21:00 UTC on 13 February 2018

(3 hours prior to the speed-up stage, Fig. 7a). The mountain wave propagated toward the northeastern direction (parallel to the TMR) associated with the interactions between the prevailing west-southwesterly winds and topography (lee wave in Fig. 7a). Stronger downdrafts and updrafts were characterized by positive and negative phases stronger than 3 m s$^{-1}$ at the

DGW, BKC and GWW sites, and the phase lines were parallel to the orientation of the TMR.

Subsequently, in the speed-up stage, the mountain wave structure significantly changed at 03:00

UTC on 14 February 2018. The wavelength became longer, but the wave was still parallel to the

TMR and the northeastern coast (Fig. 7b).

[Figure]

**Figure 7.** Horizontal distribution of the vertical velocity (m s$^{-1}$, color shading) and horizontal winds (vectors) at 2
km MSL from LDAPS in the domain corresponding to Fig. 1a at (a) 21:00 UTC on 13 Feb. 2018 and (b) 03:00 UTC
on 14 Feb. 2018. The locations of the scanning Doppler lidar and sounding sites are denoted by asterisks and squares,
respectively.

The cross section of the potential temperature (thick solid line in Fig. 8) and streamwise velocity (colors in Fig. 8) perpendicular to the orientation of the TMR demonstrated the mountain wave characteristics on the leeward side (between the DGW and GWW sites) at 21:00 UTC on

February 2018 (Fig. 8a). During this time period, a relatively strong streamwise velocity occurred only near the downslope of the TMR (~128.78ºE, ~1 km MSL) and coincided with the stronger downdraft. Weaker streamwise velocity (<4 m s$^{-1}$) appeared near the GWW site in the coastal area. However, the potential temperature pattern in the speed-up stage was characterized by a longer wavelength with a higher amplitude of the mountain wave (Fig. 8b), which is consistent with the vertical velocity field (Fig. 7b). A stronger wind exceeding 30 m s$^{-1}$ (shaded orange colors in Fig. 8b) stemmed from higher altitudes as the jet stream approached the Korean

Peninsula at this time. It was clear that strong upper winds propagated toward the lower layer and intensified the wind speed at 03:00 UTC. The range of Froude number ($F_r = U/NH$) related to the environmental winds ($U$) at the DGW sounding site was estimated to be approximately 0.55–

0.89. These Froude numbers were calculated by using dry and saturated Brunt-Vaisala frequency ($N$) with different representative terrain heights ($H$) from 1000 to 2000 m MSL (the average elevation in the TMR is ~1200 m). These upstream environmental conditions and characteristics of winds were similar to those from previous numerical studies on the northeastern coast of

Korea, and the stronger streamwise velocity extended from the upper to lower layers (exceeding

~36 m s$^{-1}$) coincident with the downdraft at downslope of the TMR (~128.78ºE, ~1 km MSL).

Along with this, surface wind was intensified exceeding ~12 m s$^{-1}$ near the surface at the GWW

site associated with the downslope wind. Note that the magnitudes of streamwise velocity are consistent with the fluctuations in the surface wind speed observed from the AWS (wind speed in Fig. 6). In this event, the impacts of the ocean on the temperature over the land would be small.

The analysis of surface sea temperature (from ERA5) indicates consistent values of approximately 6.85ºC offshore of the northeastern coast of Korea during the entire research period (not shown), however, the temperature fluctuates from ~2ºC to ~9ºC at the GWW site (cf.

Fig. 6) associated with the LPS at the same time.

[Figure]

**Figure 8.** Vertical cross section of the LDAPS potential temperature (K, contours), streamwise velocity (m s$^{-1}$, color shading), and airflow (vectors) along the black lines in Fig. 7a at (a) 21:00 UTC on 13 Feb. 2018 and (b) 03:00 UTC on 14 Feb. 2018. The red line in the lower portion indicates the topography along the black line in Fig. 7. Note that the color bars are different from the Fig. 6.

**4.1.2 Possible mechanisms of a dramatic acceleration**

The winds could usually be accelerated by the PGF between the two different locations, as the stronger wind usually occurred at the site where lower pressure was located. Therefore, the DGW site was selected as the upstream location from the GWW site, and the differences in their surface temperatures and sea level pressures were analyzed. A relatively warm environment was present om the leeward side of the TMR, and the temperature difference between the DGW and GWW sites suddenly increased from ~7°C at the beginning of the research period to ~8.5°C after 00:00 UTC on 14 February (Fig. 9a). The expected temperature difference between the two sites is approximately 6.9°C (adiabatic cooling rate for 0.7 km height difference) when adiabatic heating is assumed. The sea level pressure also decreased from ~−1 hPa to −4 hPa when the temperature increased. The observed wind speed at the GWW site showed no obvious changes in the beginning. However, the wind speed significantly increased just ~1 hour after the sea level pressure decreased and the temperature increased. This result revealed that the changes in wind speed are possibly related to the fluctuations in temperature and pressure. To clarify the effect of the pressure gradient on the wind speed at the DGW site, the local accelerations between the two sites could be approximated based on the horizontal momentum equation expressed as

$$\underbrace{\frac{\partial u}{\partial t}}_{A} = \underbrace{-u\frac{\partial u}{\partial x}}_{B} \underbrace{-\frac{1}{\rho}\frac{\partial P}{\partial x}}_{C} + \underbrace{fv}_{D} + \underbrace{\frac{C_d W_s u}{H}}_{E}. \tag{10}$$

In equation (10), Term A is the change in the $u$ component with time and also corresponds to the wind accelerations along the west-east direction, and Term B is the advective acceleration amount relative to the distance ($x$) between these two selected sites. Only the $u$ component was considered in this study since the $v$ and $w$ components could be neglected because the environmental winds were mostly composed of westerlies (Yu et al., 2020). The PGF was indicated by Term C, where $\rho$ is the air density and $P$ is the sea level pressure. Coriolis acceleration and friction were indicated by Term D and Term E, respectively, where $C_d$, $W_s$, and $H$ in Term E are the drag coefficient, wind speed and boundary layer height, respectively.

The value of the drag coefficient would most likely be a unitless constant based on Stull (1988)

and was set as ~3.9 $\times$ $10^{-3}$ in this study. The representative height of $H$ used in this study was

150 m (MSL) according to the mean boundary layer height from GWW, and a height of 1500 m (MSL) was observed from the DGW sounding observations during 12:00 UTC on 13 and 00:00

UTC on 15 February 2018 (not shown).

Basically, the wind accelerations (i.e., Term A) that are derived from equation (10) by adding terms from B to E are in good agreement with the fluctuations in wind speed at the GWW site (Fig. 9b). A relatively weak wind speed occurred in the beginning and coincided with negative and weak accelerations. Consequently, the wind speed rapidly increased at the GWW site in the speed-up stage (i.e., shaded area in Fig. 9) associated with the increased and positive accelerations (i.e., Term A). Furthermore, the contributions of Terms B–E to Term A could also be evaluated individually by calculating each term. The PGF (Term C) dominated the changes in Term A with almost the same magnitudes during the entire research period as shown in Fig. 9c. In the beginning, advective acceleration (Term B) could provide slight positive contribution to Term A, while the PGF term was negative. However, both Term B and friction (Term E) gave negative feedback to Term A in the speed-up stage. Coriolis acceleration (Term D) always exhibited an almost zero acceleration to Term A in such small-scale wind patterns (~10 km distances between these two sites and time periods of a few hours). The results suggested that the PGF would be the main factor dominating the changes in wind speed at the GWW site on the leeward side of the

TMR.

[Figure]

**Figure 9.** (a) Time series of wind speed (m s⁻¹, black line) observed from the AWS at the GWW site and the differences in sea level pressure (hPa, blue line) and temperature (ºC, red line) between the GWW and DGW sites from 12:00 UTC on 13 Feb. to 12:00 UTC on 14 Feb. 2018. (b) Time series of the u component acceleration ($10^{-3}$ m s⁻², Term A, black dashed line) estimated from the horizontal momentum equation [eq. (10)] between the GWW and DGW sites. (c) Time series of the u component acceleration (Term A, black dashed line), advective acceleration (Term B, red line), PGF (Term C, blue line), Coriolis acceleration (Term D, gray line), and friction (Term E, green line). The time period with accelerating wind speed is also highlighted by light yellow shading (i.e., speed-up stage).

Since the gusty wind (the wind speed suddenly increased from ~3 to 12 m s⁻¹ at the GWW site, cf. Fig. 9a) was mainly explained by the PGF and this result was also consistent with the fluctuations in the sea level pressure from the AWS observations at the GWW site (cf. Fig. 6b), the observed station pressure at the GWW dropped deeply during the speed-up stage. To understand the possible causes of the relatively low pressure occurring on the leeward side of the

TMR, more detailed analysis is needed.

To evaluate the contributions of pressure and temperature from large-scale weather systems, average values of sea level pressure and potential temperature were calculated from selected 24

AWSs (Fig. 1b) to represent the contributions of the passing LPS. The elevations of the selected

AWSs must be higher than the GWW site to avoid the effects of adiabatic warming along the northeastern coast. The time series of average sea level pressure and average potential temperature are shown in Fig. 10 with the speed-up stage indicated by shading. In the speed-up stage, the average sea level pressure decreased ~3 hPa (from ~1015 hPa to 1012 hPa), and the average potential temperature increased ~3K (from ~279.5K to 282.5K) (Fig. 10). This variation (contributions from large-scale weather systems, i.e., from the LPS) is similar to sea level pressure (~−3 hPa: from approximately −1 hPa to −4 hPa in Fig. 9a) and temperature (~2.5°C:

from approximately 6 °C to 8.5 °C) difference between the DGW and GWW site. Therefore, the coupled effect of adiabatic warming and the passing LPS is probably the main factor that induced the extreme winds on the leeward side of the TMR.

[Figure]

**Figure 10.** Time series of average sea level pressure (SP, blue line, unit: hPa) and average potential temperature ($\theta$, red line, unit: K) over the 24 AWS stations. from 12:00 UTC on 13 Feb. to 12:00 UTC on 14 Feb. 2018. The time period with accelerating wind speed is also highlighted by light yellow shading (i.e., speed-up stage).

**4.2 The winds in mountainous areas**

**4.2.1 Persistent strong winds in mountainous areas**

The combination of the LPS and HPS provided a large-scale environmental wind favorable for westerly winds over the mountainous area. According to the DGW QVP from observations (Fig. 11a), the wind speed ranged from ~12 to 36 m s$^{-1}$ at the low-level layers (~900 to 1800 m MSL) during 12:00 UTC on 13 February to 12:00 UTC on 14 February 2018. After this time period, the wind decayed so quickly that it became nearly calm associated with the approaching HPS (Fig. 2e). The surface wind fluctuated in the range of 7 m s$^{-1}$ to 12 m s$^{-1}$ with a periodicity of 6 hours at the DGW site, similar to the pattern in the lidar QVP (Fig. 11b). These characteristics were quite different from the AWS and lidar observations on the leeward side of the mountains (for example, the GWW site). Unlike the coastal site, the strong wind was sustained for a day in the mountainous area. In particular, there were persistent westerly winds at all altitudes over the mountainous area, and the winds were enhanced, especially in some local areas (i.e., the DGW site). However, the wind direction was quite variable from southerly to westerly on the leeward side of the TMR (GWU or GWW site). Significant strong winds were measured at the DGW site above 1000 m MSL on 13 February (Fig. 11), and the wind was weak at the GWU site (Fig. 6). Although the wind strengthened at the GWU and GWW sites at ~02:00 UTC on 14 February, the low-level or surface winds became slightly weaker at the DGW site. This pattern is consistent with the wind fields from the sounding observations at the DGW site (Figs. 4a and 4b).

The AWS observations at the DGW site demonstrated sustained strong westerlies (~10 m s$^{-1}$) with periodic fluctuations from 12:00 UTC on 13 February to 12:00 UTC on 14 February 2018 (Fig. 11b). Although the wind speed fluctuated periodically, no periodicity was shown in the station pressure or temperature. Instead, the station pressure monotonically dropped from ~925 hPa at 12:00 UTC on 13 February to ~920 hPa at 05:00 UTC on 14 February and then increased back to ~925 hPa at 12:00 UTC on 14 February 2018. The temperature showed a nearly opposite trend to that of the station pressure. The temperature was nearly steady until 22:00 UTC and then increased from approximately -4 to 3ºC at 07:00 UTC on 14 February 2018. Although the movement of the LPS affected the changes in the station pressure and temperature at the DGW

site, the changes in the wind speed had no clear relation with the station pressure or temperature.

[Figure]

**Figure 11.** Same as Figure 6, but for DGW site.

**4.2.2 Possible mechanisms of persistent strong winds**

To document the possible mechanisms of sustained strong winds occurring at the DGW site over the mountainous area, differences in temperature and pressure were analyzed in detail. A western surface station (YPO site in Fig. 1b) was selected to calculate the temperature and pressure differences with the DGW site. Fig. 12a reveals that the fluctuations in pressure differences (blue line in Fig. 12a) had an almost negative relation (opposite phase) with the fluctuations in wind speed (black line in Fig. 12a) at the DGW site. Furthermore, the wind speed gently decreased with periodicity (wavelength of approximately 6 hours).This result provided a clue that the pressure gradient likely dominated the wind speed in this local area. Compared to the leeward side of the mountains at the GWW site (Fig. 9), negative values of the temperature differences (minimum of −1.3ºC) were calculated in the mountainous area and even became smaller (−0.5ºC) after 12:00 UTC on 14 February. Thus, the differences in pressure seemed to affect the wind speed patterns, and the fluctuations in wind speed were less related to the differences in temperature between these two sites. The periodic characteristics of the surface wind may have been linked to nonlinear dynamics, such as gap flow and gravity wave mechanisms (Shun et al., 2003).

The acceleration of wind speed at the DGW site can also be estimated by equation (10). Most of the estimated Term A and wind speed were also in a good agreement except for a short time period (Fig. 12b). Basically, the wind speed increased (decreased) when the estimated acceleration (i.e., Term A) was positive (negative). To understand the main contributor that dominates such strong local winds in this area, a detailed budget analysis of the momentum equation was performed (Fig. 12c). The PGF (Term C) was the most important factor for the estimated acceleration, which means that the PGF could possibly determine the changes in the wind speed at the DGW site. The advective acceleration was relatively small. The Coriolis force and friction had no clear impacts on the acceleration (Term A).

[Figure]

Surface Observation (AWS)    Station:100 DGW - 318 YPO

**Figure 12.** Same as Fig. 9, but for the DGW and YPO sites, and the y-axis indicates the wind speed at DGW site.

The above results show that the PGF is the main factor accelerating wind speed, but temperature is not a critical factor changing the PGF over the mountainous area. To determine the possible factors that contribute to the PGF, a more detailed analysis of horizontal winds was performed with WISSDOM synthesis. Fig. 13 demonstrates the fine-scale wind fields at 800 m MSL (near the surface in the studied domain). At this height, a unique topographic feature was explored, as it occurred over a relatively wide (narrow) area on the western (eastern) side along the valley. This channel-like feature is marked by the area between two thin dashed lines in Fig. 13 for emphasis. Four periods (00:00 UTC on 13 February, 00:00 UTC on 14 February, 12:00

UTC on 14 February, and 00:00 UTC on 15 February 2018) were selected to investigate the changes in wind patterns in this channel along the valley. The prevailing wind was westerly with a slight deflection near the center of the domain and the eastern side of the valley, while the LPS

approached Korea (Figs 13a, 13b, and 13c). Nevertheless, a relatively weak wind (~6 m s$^{-1}$)

always existed in the center of the domain near the MHS lidar site (wide segment of the valley)

and a stronger wind (14 m s$^{-1}$) was observed near the DGW site (narrow segment of the valley).

The wind speed decreased and nearly became calm after the LPS moved away from Korea (Fig.

13d).

[Figure]

**Figure 13.** Horizontal distribution of the wind speed (m s$^{-1}$, color shading) at 800 m (MSL) retrieved in the WISSDOM domain at (a) 00:00 UTC on 13 Feb., (b), 00:00 UTC on 14 Feb., (c) 12:00 UTC on 14 Feb., and (d) 00:00 UTC on 15 Feb. 2018. The black dashed lines mark the area of the channel to calculate the average wind speed and channel width as shown in Fig. 14. The rectangular box indicates the average area in the vertical cross sections along the valley (A-A'). Topographic features are indicated by the gray shading and contours. Locations of the scanning Doppler lidar sites are denoted by asterisks.

The relations between the topography, average wind speed (thick colored lines in Fig. 14) and channel width (thick black line in Fig. 14) along the valley at 800 m MSL were calculated in two time periods when the LPS was approaching (before 12:00 UTC on 14 February 2018) and leaving (after 12:00 UTC on 14 February 2018). The channel width was approximately 2 km at x = 0 km to 3 km (western side) and became wider (~5.5 km) at x = 3 km to 6.5 km. The channel width then decreased significantly to nearly 0 km at x = 6.5 km to 9.5 km.

When the LPS was approaching (average wind speed in red line and range of minimum and maximum wind speed in shading in Fig. 14), the average wind speed increased from ~10 m s$^{-1}$

to ~14 m s$^{-1}$, which was coincident with the change in channel width from ~5.5 km to 0 km along the valley. When the LPS was leaving (blue line and shading), the average wind speed increased from ~5 m s$^{-1}$ to ~ 7 m s$^{-1}$ in the narrow segment. There was a similar increase in wind speed of

~40% in these two stages, and this result also reflected that the wind was indeed accelerated by the channeling effect in this local area. However, the maximum wind speed was larger than 24 m s$^{-1}$ near the narrowest segment of the valley when the LPS was approaching and was only 12 m s$^{-1}$ when the LPS was leaving. The maximum wind speed was amplified significantly (~10 m s$^{-1}$

more than average) in the narrow segment along the valley when the westerly winds were stronger. In contrast, the wind speed was amplified by only 6 m s$^{-1}$ when prevailing winds became weaker. This analysis reveals that the channeling effect may play an important role in dominating the spatial distribution of wind speed with the valley.

**Figure 14.** Average wind speed and its range along the valley corresponding to the area indicated by the dashed lines in Fig. 13 at two times: 12:00 UTC on 13 Feb. to 12:00 UTC on 14 Feb. (red line and shading) and 12:00 UTC on 14 Feb. to 00:00 UTC on 15 Feb. (blue line and shading) 2018. The red and blue shading show the maximum and minimum values along the valley for the two times. The average channel width along the valley is plotted by a thick black line.

Fig. 15 shows the mean vertical structures of wind speed, airflow, and topographic features from each cross section along the boxes in Fig. 13. The boxes were set on our main focus area from wider to narrower segments along the valley and parallel to the environmental wind direction (westerly). These analyses allow us to investigate detailed airflow features from near the surface to higher altitudes and their interactions with topography. The four time periods were 12:00 UTC on 13 February 2018, 00:00 UTC and 12:00 UTC on 14 February, and 00:00 UTC on 15 February 2018. The mean vertical structures in the first three periods (when the LPS was approaching) revealed similar characteristics: uniform and stronger westerly winds (larger than ~18 m s$^{-1}$) in the layers above 1 km MSL. In contrast, the airflow had more significant variability in the layers near the surface. In the layers below 1 km MSL, the westerly winds were lifted upslope and became downdrafts behind the mountain crests. In the three time periods, the wind speed was quite weak near the MHS site and was strong near the DGW site, which are coincident with the relatively wide and narrow segments in the valley, respectively. In particular, the high wind speed area was only between x = ~6.5 km and 9.5 km (i.e., the narrowest segment of the valley). The winds became more uniform and weaker in the upper layers and near the surface when the LPS moved away from Korea at 00:00 UTC on 15 February 2018 (Fig. 15d).

[Figure]

**Figure 15.** Average vertical cross section of the WISSDOM-derived wind speed (m s$^{-1}$, color shading) and wind vectors (combined cross barrier flow and threefold vertical velocity) at four time periods (a) 12:00 UTC on 13 Feb.

2018, (b) 00:00 UTC on 14 Feb. 2018, (c) 12:00 UTC on 14 Feb. 2018, and (d) 00:00 UTC on 15 Feb. 2018. The area of the cross section is shown by the black box in Fig. 13. The black line in the lower portion indicates the topography along the box.

Because the winds manifested clear variations only near the surface layers, the mean vertical structures of wind speed and directions could be further averaged below 1 km MSL. Fig. 16

shows the continuous time series of the averaged wind field during the entire period with the same x axis as that in Fig. 15. The results demonstrate that the winds near the surface layers were accelerated in the narrow segment between x = ~6.5 km and 9.5 km for sufficiently strong westerly winds (before 00:00 UTC on 14 February). This characteristic is similar to the gap wind or channeling effect from previous simulation and observational studies (Overland and Walter,

1981; Neiman et al., 2006; Heinemann, 2018). Consequently, a relatively weak channeling effect induced weaker winds in the narrow segment of the valley during 00:00–15:00 UTC on 14

February 2018 because the environmental winds became weaker. Finally, the channeling effect no longer exited when the environmental winds became calm after 15:00 UTC. The wind might accelerate when it blows from wider to narrower segments of the valley due to the PGF, as indicated by Bernoulli's Law, i.e., the pressure decreases when the flow speed increases and vice versa. Observational analysis reveals a relatively low pressure in the narrow segment of the valley, and thus, the PGF would locally dominate the airflow acceleration over the mountainous area.

[Figure]

**Figure 16.** Temporal variation in the average wind speed (m s$^{-1}$, color shading) and the horizontal winds (wind barbs) from WISSDOM derived in the valley from 00:00 UTC on 13 Feb. to 00:00 UTC on 15 Feb. 2018. The low-level winds (below 1 km MSL) within the black boxes in Fig. 15 were averaged in a direction normal to the orientation of the boxes. The black line in the lower portion indicates the average topography along the boxes.

**5. Conclusion**

This study uses Doppler lidar, wind profiler, sounding, surface observation, global reanalysis (ERA5) and local reanalysis (LDAPS) datasets to examine an extreme wind event during ICE-POP 2018. Detailed characteristics of wind fields and possible mechanisms during the passage of a low-pressure system (LPS) over the northern part of the Korean Peninsula on 13–15 February 2018 were explored. Although the wind speed in South Korea generally increased when the LPS was approaching, the winds comprised more significant gusty winds along the downslope and on the leeward side of the Taebeak Mountain Range (TMR). In contrast, the wind speed was persistently strong in several local areas over the TMR. Conspicuous gradients in wind speed patterns existed only between the mountainous areas and the leeward side of the mountain range. Moreover, the wind speed decreased synchronously after the LPS moved away from Korea.

From the sounding observations, low-level environmental winds revealed high variability from the mountainous area to the leeward side of the mountains. The wind direction was mostly westerly associated with the LPS, and the wind speed was persistently strong (~10 m s$^{-1}$) at the DGW site (i.e., mountainous area) during the research period. However, the wind speed on the leeward side (GWW) clearly changed from relatively weak to stronger. The winds then become nearly calm both in the mountainous area and on the leeward side of the mountain range after the LPS moved away from the Korean Peninsula. In addition, upstream inversion layers (at the ~850 hPa level) were also detected by sounding observations at the DGW site, while strong winds occurred on the leeward side of the mountains.

On the leeward side of the mountains, the surface wind speed dramatically increased (from ~3 to 12 m s$^{-1}$) at the GWW site during the research period. The surface temperature increased and station pressure decreased, and the fluctuations in temperature and pressure showed a significant time lag with wind speed changes. In addition, the strong winds were well depicted along the downslope and the leeward side of the TMR from the LDAPS reanalysis data. This result is similar to those from previous numerical studies in Korea, and the development of strong downslope winds is related to mountain waves and hydraulic jumps. In the mountainous area, persistent strong surface winds were observed at the DGW site when the LPS was approaching.

The surface wind has no clear relationship with the station pressure or temperature during the research period.

The sea level pressure and temperature differences between the mountainous station at the

DGW site and the leeward station at the GWW site demonstrate that the wind speed suddenly increased with increasing temperature (from ~6ºC to 8.5ºC) and decreasing sea level pressure (from ~−1 hPa to −4 hPa) in the speed-up stage. The estimated wind accelerations [Term A in eq.

(10)] are in good agreement with the observed wind speed, which are mainly contributed by the

PGF [Term C in eq. (10)]. In the speed-up stage, the average sea level pressure and potential temperature in the AWS observations show fluctuations of approximately −3 hPa and +3K when the LPS passed over. The differences in the sea level pressure (~−3 hPa) and temperature (~2.5ºC)

between the DGW and GWW sites were almost equal to the contributions from large-scale weather systems. The results indicate that adiabatic warming coupled with the LPS plays an important role in reducing the surface pressure and those winds are accelerated by the PGF on the leeward side of the TMR.

The sea level pressure differences between the YPO and DGW stations show almost negative relations with the fluctuations in surface wind speed. In contrast, the temperature differences are small (between −0.5ºC and 1.2ºC) with no clear relations with the fluctuations in surface wind at the DGW site. Although the temperature has no clear relation with the strong wind, estimated wind accelerations [Term A in eq. (10)] are in good agreement with the observed surface wind speeds. This means that the PGF is still the main contributor to the wind acceleration at the DGW site. The 3D winds derived from WISSDOM synthesis also reveal that the wind speed at the DGW site (narrow segment in the valley) was always stronger than that at the YPO

site (wider segment in the valley). In addition, the channeling effect was amplified to effectively accelerate the winds at the DGW site when the westerly winds were stronger due to the approaching LPS. Thus, the channeling effect is a possible mechanism dominating the wind acceleration in the mountainous area.

In this study, observationally based evidence shows that different mechanisms are important for determining the strength and persistence of orographically strong winds in the same underlying LPS under clear air conditions. In the future, high-resolution numerical modeling analysis will be performed for all strong wind events during ICE-POP 2018 because detailed thermodynamic information is desired to provide more complete descriptions about the distribution of potential temperature across the mountainous area. The kinematic and thermodynamic information from the simulations will be important indicators to further investigate the existence of mountain waves, including hydraulic jumps, wave breaking, and partial reflection for the generation of the downslope windstorms. More cases will be included to provide comprehensive explanations of the strong downslope wind in the northeastern mountainous part of South Korea. More importantly, we aim to extend our understanding of the variability in winds around terrain on a very fine-scale even in different seasons.

*Author contributions.* This work was made possible by contribution from all authors.
Conceptualization, CLT, GWL, JHK ; methodology, CLT, YCL, YHL, JHK, and KK; software,
CLT, YHL, and KK; validation, KK, YHL, and GWL; formal analysis, CLT, and JHK;
investigation, CLT, GWL, and YHL; writing—original draft preparation, CLT; writing—review
and editing, GWL, JHK, YCL and YHL; visualization, CLT; supervision, GWL, and YHL;
funding acquisition, GWL, YHL, and JHK. All authors have read and agreed to the published
version of the manuscript.

*Competing interests.* The authors declare that they have no conflict of interest.

*Special issue statement.* This article is part of the special issue "Winter weather research in
complex terrain during ICE-POP 2018 (International Collaborative Experiments for

PyeongChang 2018 Olympic and Paralympic winter games) (ACP/AMT/GMD inter-journal SI)".

It is not associated with a conference.

*Acknowledgments.* This work was supported by Civil-Military Technology Cooperation Program funded by the Korea Meteorological Administration and Defense Acquisition Program

Administration. No. 17-CM-SS-23, KMA2017-04210, [Project Name : Development of fusion technology for Radar wind profiler] and was funded by the Korea Meteorological Administration

Research and Development Program under Grant KMI2020-00910. The authors greatly appreciate the participants in the World Weather Research Programme Research Development

Project and Forecast Demonstration Project, International Collaborative Experiments for

Pyeongchang 2018 Olympic and Paralympic winter games (ICE-POP 2018) hosted by Korea

Meteorological Administration (KMA). The Doppler lidars were deployed by the National

Institute of Meteorological Sciences (NIMS), KMA and Environment Climate Change Canada (ECCC). We would like to thank many researchers and students (Byung-Chul Choi, Kwang-Deuk

Ahn, Namwon Kim, and Seung-bo Choi at KMA, and Choeng-lyong Lee, Daejin Yeom, Kyuhee

Shin, DaeHyung Lee, Su-jeong Cho, SeungWoo Baek, Hong-Mok Park, Geunsu Lyu, Eunbi

Jeong, Heesang Yoo, Youn Choi, Bo-Young Ye, and Soohyun Kwon at Kyungpook National

University) who collected data during the ICE-POP 2018 period.